# High-resolution genome-wide mapping of chromosome-arm-scale truncations induced by CRISPR–Cas9 editing

Nathan H. Lazar, Safiye Celik, Lu Chen, Marta M. Fay, Jonathan C. Irish, James Jensen, Conor A. Tillinghast, John Urbanik, William P. Bone, Christopher C. Gibson & Imran S. Haque ✉

Clustered regularly interspaced short palindromic repeats (CRISPR)–CRISPR-associated protein 9 (Cas9) is a powerful tool for introducing targeted mutations in DNA, but recent studies have shown that it can have unintended effects such as structural changes. However, these studies have not yet looked genome wide or across data types. Here we performed a phenotypic CRISPR–Cas9 scan targeting 17,065 genes in primary human cells, revealing a 'proximity bias' in which CRISPR knockouts show unexpected similarities to unrelated genes on the same chromosome arm. This bias was found to be consistent across cell types, laboratories, Cas9 delivery methods and assay modalities, and the data suggest that it is caused by telomeric truncations of chromosome arms, with cell cycle and apoptotic pathways playing a mediating role. Additionally, a simple correction is demonstrated to mitigate this pervasive bias while preserving biological relationships. This previously uncharacterized effect has implications for functional genomic studies using CRISPR–Cas9, with applications in discovery biology, drug-target identification, cell therapies and genetic therapeutics.

Clustered regularly interspaced short palindromic repeats (CRISPR)–CRISPR-associated protein 9 (Cas9)-based methods are powerful genome editing tools with applications in in vitro discovery biology, ex vivo editing for cell therapies and in vivo editing for genetic therapeutics[1]. Cas9 is programmably scalable and relatively specific compared with earlier technologies such as zinc finger nucleases, TALENs, and small interfering RNAs[2,3]. However, CRISPR–Cas9-based editing is known to have off-target activity and undesired on-target changes, such as kilobase-scale deletions[4–6], chromosome truncation[7–10] and complex rearrangements[5,6]. Profiling these effects systematically is crucial for discovery and therapeutic development but is costly and labor intensive with existing molecular or sequencing-based methods.

Pooled CRISPR–Cas9 knockout screens have been widely used to identify essential genes in tumor-derived cell lines[11–13]. These studies

have reported associations between copy number variants (CNVs) and chromosomal instability at the target site after CRISPR–Cas9 editing and introduced methods to help correct for those effects[14–18]. However, they have not explored the effects of CRISPR–Cas9-induced chromosomal changes that are unrelated to CNVs nor have they explored these effects in primary cell types or with endpoints other than essentiality.

Cellular morphological profiling, or 'phenomics', is an emerging technology for high-dimensional phenotyping that offers a powerful alternative to transcriptomic or proteomic assays[19]. Measuring cellular morphology, a holistic functional endpoint of the cellular state, generates high-dimensional single-cell data at a lower cost than molecular methods such as single-cell RNA sequencing (scRNA-seq).

We applied phenomics to systematically profile CRISPR–Cas9 knockouts in primary human cells, targeting over 17,000 genes with

Recursion, Salt Lake City, UT, USA. ✉e-mail: Imran.Haque@recursionpharma.com

more than 100,000 CRISPR guides. A proprietary deep-learning model encoded six-stain fluorescent images of plated cells[19], producing a single 'gene vector' representing the phenotype of each perturbed gene. Cosine similarity between gene vectors acted as a pairwise measure of phenotypic similarity between knockouts, recapitulating known and novel biological relationships, including protein complexes and annotated pathways, and can be extended to assess similarity among a broad range of cellular perturbations, including genetic, large-molecule and small-molecule treatments[20]. In this Article, we report the observation of 'proximity bias', where CRISPR knockout phenotypes are systematically more similar to unrelated genomically proximal genes located on the same chromosome arm. This effect is found to be general across laboratories, cell types and Cas9 delivery mechanisms and is dependent on nuclease activity. Also, patterns of proximity bias reflect differences between reference genomes and true chromosomal structure, including large-scale structural variants. Molecular investigation with bulk and single-cell transcriptomic analysis supports large-scale chromosomal truncation as the driving mechanism. Additionally, we reanalyzed the Cancer Dependency Map (DepMap) genome-wide CRISPR–Cas9 screens[21] in cancer cell lines to confirm the impact of proximity bias on target discovery, propose potential mediators and show that this effect persists even when controlling for cell-line specific CNVs. Finally, we show that an arm-based normalization of gene-level features largely corrects for this bias without affecting the recovery of known biological relationships.

## Results

### Genome-wide profiling recapitulates known biology

To produce a genome-wide 'map' of pairwise phenotypic similarity between gene knockouts, we performed a phenomics screen using CRISPR–Cas9 to knock out 17,065 genes in primary human umbilical vein endothelial cells (HUVEC) with 101,029 guides (typically six guides per gene and 24 replicates per guide) leveraging a highly automated robotic workflow (rxrx3 dataset)[22].

To validate phenomics, we computed a complementary similarity map using the cpg0016 dataset from the Joint Undertaking in Morphological Profiling–Cell Painting consortium[23]. The key differences between rxrx3 and cpg0016 include gene sets, cell types and Cas9/guide delivery protocols. Cpg0016 profiles fewer genes ($n = 7,975$) with fewer samples ($n = 4$ guides and five replicates per guide), screens the U2OS osteosarcoma cell line and uses lentiviral Cas9 delivery with lipofection of guide RNA pools. For rxrx3, we applied a proprietary deep-learning model to extract features; for cpg0016, we used the CellProfiler-derived features[24] provided in the Joint Undertaking in Morphological Profiling–Cell Painting source data. For both datasets, the gene vectors were aggregated to build a genome-wide 'map' to compare phenotypic similarities using cosine similarity (Fig. 1a).

We evaluated the ability of rxrx3 and cpg0016-based maps to recapitulate known biology in both a targeted and a broad sense. Targeted examination of genes in well-studied pathways showed that gene–gene similarities recapitulate both biology that is highly conserved across cell types (for example, microtubule, proteasome and autophagy genes), as well as therapeutically relevant pathways including Janus kinase (JAK)/signal transducer of activation (STAT), transforming growth factor (TGF)-beta and insulin receptor (Fig. 1b and Extended Data Fig. 1). Despite the methodological differences between datasets, large-scale benchmarking[20] of both datasets shows substantial recall of known annotations drawn from public datasets, including Reactome[25], HuMap[26] and CORUM[27] (Fig. 1c).

### Knockouts show increased similarity within chromosome arms

Upon the generation of the rxrx3 full-genome knockout data, we noticed a curious bias: the distribution of cosine similarities for gene pairs on the same chromosome was shifted relative to gene pairs on different chromosomes (Extended Data Fig. 2a). Visualizing the full genome-wide dataset ordered by genomic coordinate showed a striking structure in which knockouts of genomically proximal genes on the same chromosome arm were systematically more phenotypically similar to one another than distal pairs (Fig. 1d,e). To test whether proximity bias was an artifact of our experimental set up (laboratory protocol, Cas9 and guide delivery system, HUVEC cell type, computational image analysis and featurization scheme and so on), we performed the same visualization in cpg0016 and found a very similar effect both visually in the genome-wide heat map (Fig. 1d,e) and in intra- versus interchromosomal similarity distributions (Extended Data Fig. 2a).

As proximity blocks appeared to correlate with chromosomal structure, we wondered whether proximity blocks would also reflect nonreference structure in genomically abnormal cells. The U2OS line used in cpg0016 is known to be karyotypically abnormal and heterogeneous, with different clones exhibiting distinct genotypes[28]. DepMap[11] cataloged a fusion between *RAD50* on chromosome 5q and *ZNF536* on chromosome 19q in this cell line, and examination of the cpg0016 map shows a clear block of interchromosomal proximity bias between 5q and 19q that closely recapitulates the boundaries of the annotated fusion (Fig. 1f).

Finally, we quantified the proximity bias effect by estimating, for each chromosome arm, the probability of a within-arm relationship displaying a higher cosine similarity than a between-arm relationship using a nonparametric Brunner–Munzel test[29]. This metric is both comparable across maps that may have different numbers of tested genes and flexible enough to be used to quantify the bias encoded in an entire map, within each chromosome arm or at the gene level (by restricting to relationships involving that arm or gene). At the full-map level, the probability of a within-arm relationship being ranked above a between-arm relationship was 0.71 for the rxrx3 dataset and 0.72 for the cpg0016 data ($P < 1 \times 10^{-10}$). At the chromosome-arm level, both datasets show a significant effect for all chromosome arms (Fig. 1g,h).

**Fig. 1 | Heat maps of phenotypic similarity between gene knockouts recapitulate known biology as well as genomic proximity effects.**
**a**, Phenomics overview. Screening of the genetic perturbations produces images of cells from which features are extracted either using CellProfiler[24] or neural networks. The feature vectors for each pair of perturbations are related using cosine similarity (ranging from −1 opposite ('opp.') to 1 similar ('sim.')) and visualized in heat maps. **b**, A heat map of genes with diverse functions. The rows and columns are clustered on the rxrx3 data. EGFR, epidermal growth factor receptor; TGFB, transforming growth factor-beta. **c**, Recall of annotated known relationships from three databases in the most extreme 10% of similarities (two sided). A random ranking of gene–gene pairs would give a baseline value of 0.1. **d**, Full-genome heat map where each row and column represent a gene assessed in both rxrx3 (above diagonal) and cpg0016 (below diagonal) studies. Ordering genes by chromosomal position reveals the proximity bias signal along the diagonal present in both datasets with the chromosome boundaries and centromeres clearly visible. **e**, A zoom-in on chromosome 8. **f**, Juxtaposition of chromosomes 5 and 19, where the pattern of proximity bias signal reflects a chromosomal rearrangement known to be present in U2OS cells (cpg0016 data) but not HUVEC (rxrx3 data). **g**, A bar plot of proximity bias metrics (Brunner–Munzel probabilities) for each chromosome arm for the rxrx3 dataset. The values above 0.5 indicate elevated intra-arm similarity, and all chromosome arms are significant with Bonferroni correction (one-sided $P < 0.001$). **h**, A bar plot of proximity bias metrics for each chromosome arm as in **g** for the cpg0016 dataset. All chromosome arms are significant with Bonferroni correction (one-sided $P < 0.001$). In all heat maps, each row and column represent a single gene with rxrx3 data shown above the diagonal, cpg0016 data below the diagonal. Only the 7,477 genes that are present in both datasets are shown. The solid lines represent chromosome boundaries and the dashed lines indicate centromeres.

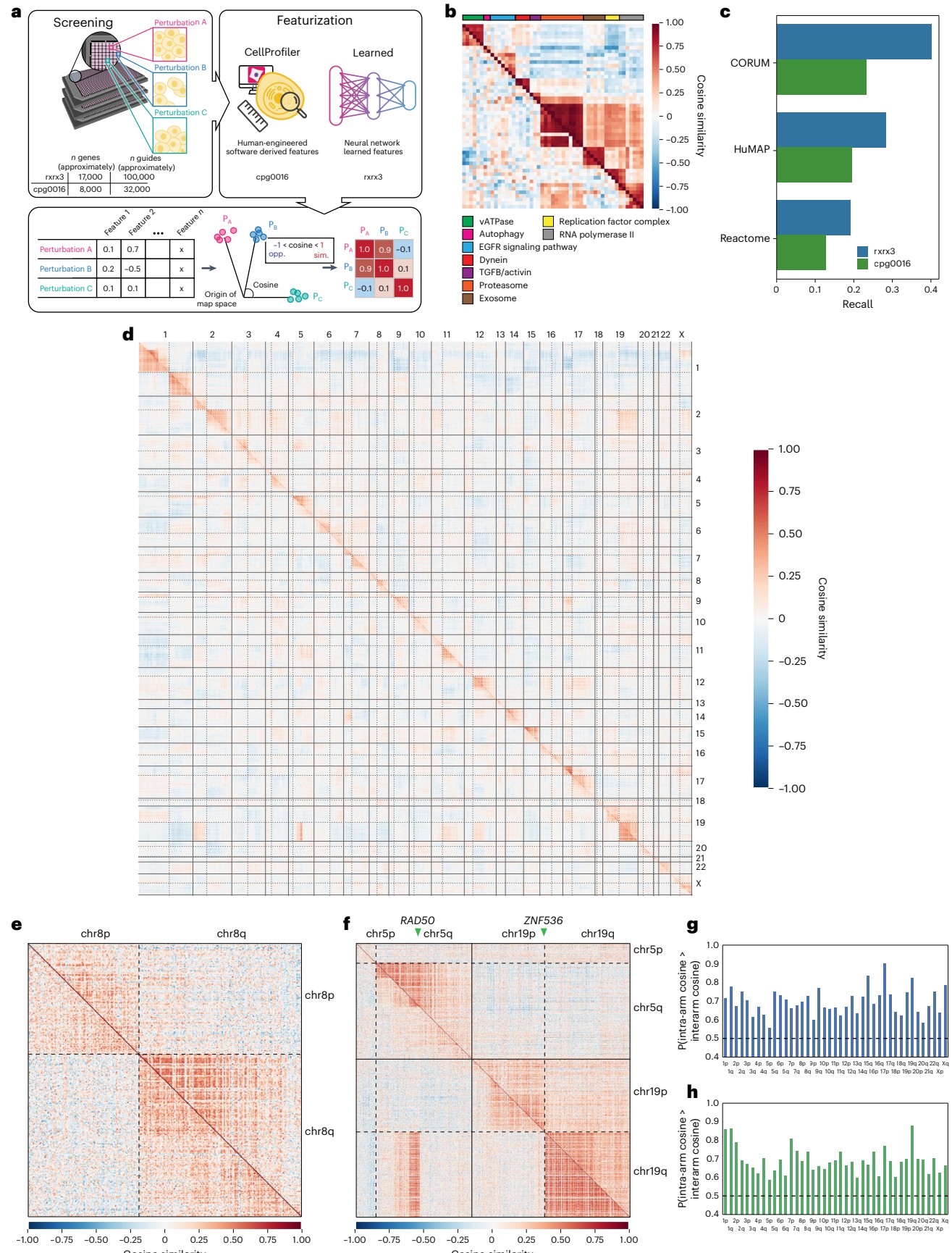

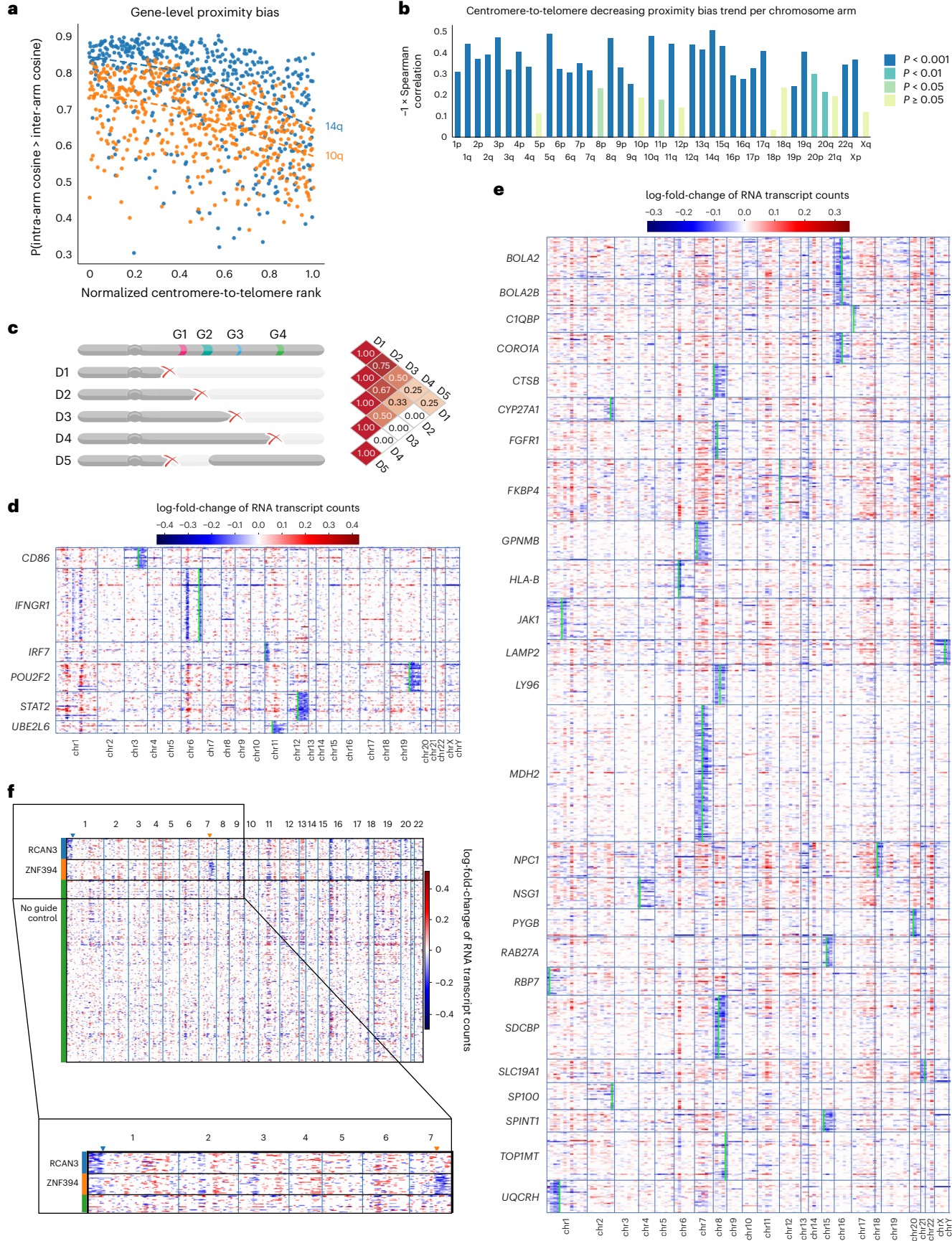

**Fig. 2 | Genome-wide phenomic measurements and transcriptomic data support a model of chromosomal truncation underlying proximity bias. a**, A scatter plot of gene-level proximity bias showing the one-sided Brunner–Munzel probability of intra-arm cosine values exceeding inter-arm cosine values versus position within the chromosome arm for two arms in rxrx3 data. A negative Spearman correlation supports the visual pattern seen in Fig. 1c of weakening proximity bias signal toward the telomeres. **b**, Spearman correlations in plots similar to Fig. 2a across all chromosome arms for the rxrx3 data. The height of the bar for each arm agrees well with the degree of fading in diagonal blocks in Fig. 1d. The colors correspond to Bonferroni-corrected *P* values. **c**, A schematic representation of how the pattern of weakening signal toward telomeres observed in chromosome-arm heat maps may arise by deletions sharing LOF of multiple key genes. D1, …, D4 represent varying-length truncations and D5 an intra-arm deletion. D1 and D2 both lose three key genes

and hence are highly similar, while D1 and D4 share only one and so may look less similar. The intra-arm deletion D5 shares only one key gene loss with D1 and zero losses with D2–D4 and, therefore, is expected to show less proximity bias. **d**, A heat map of copy number change estimates in Papalexi et al.[36] for genes resulting in proximal deletions when targeted by CRISPR–Cas9. Each row in the heat map represents a cell exhibiting deletion near the target gene as in the row label. The rows are ordered alphabetically based on target gene name. Each column represents a gene, ordered by the chromosome position. The lime bars represent the target gene positions. **e**, A heat map of copy number change estimates in Frangieh et al.[37]. **f**, A heat map of copy number change estimates in bulk RNA sequencing samples showing loss of chromosome regions telomeric to the cut site of guides targeting introns in the *RCAN3* and *ZNF394* loci. Each row in the heat map represents a treatment well. Each column represents a gene, ordered by chromosome position.

**Table 1 | Single-cell sequencing reveals widespread on-target proximal deletions from CRISPR–Cas9**

| Perturbation type | Dataset | Cell type | Total number of tested targets | Tested loss direction | Number of targets with specific loss | Number of targets with loss toward telomere | Number of targets with loss toward centromere |
|---|---|---|---|---|---|---|---|
| CRISPR–Cas9 | Papalexi | THP-1 (monocytic leukemia) | 24 | 3′ | 6 (25.0%) | 5 (20.8%) | 1 (4.2%) |
| | | | | 5′ | 1 (4.2%) | 1 (4.2%) | 0 (0.0%) |
| | Frangieh | Melanocytes (melanoma) | 237 | 3′ | 31 (13.1%) | 23 (9.7%) | 8 (3.4%) |
| | | | | 5′ | 34 (14.3%) | 20 (8.4%) | 14 (5.9%) |
| CRISPRi | Replogle | RPE1 | 2,066 | 3′ | 45 (2.2%) | 31 (1.5%) | 14 (0.7%) |
| | | | | 5′ | 45 (2.2%) | 11 (0.5%) | 34 (1.6%) |
| | Tian | Induced pluripotent stem cell-derived neurons | 177 | 3′ | 3 (1.7%) | 3 (1.7%) | 0 (0.0%) |
| | | | | 5′ | 4 (2.3%) | 2 (1.1%) | 2 (1.1%) |
| | Adamson | K562 (leukemia) | 78 | 3′ | 1 (1.3%) | 1 (1.3%) | 0 (0.0%) |
| | | | | 5′ | 2 (2.6%) | 1 (1.3%) | 1 (1.3%) |

The number and percent of target genes showing deletions specific to the target region in two CRISPR–Cas9 and three CRISPRi datasets uniformly reprocessed in scPerturb.

## Proximity bias arises from chromosome-arm truncation

Several unintended consequences of Cas9 editing have been discussed in the literature[4–10,30–34], and while the rxrx3 and cpg0016 genome-wide maps show widespread proximity bias, they do not directly nominate a mechanism. However, observing the maps in Fig. 1, we found knockouts of genes closer to a centromere often display a stronger proximity bias signal. To quantify this, we plotted the gene-level Brunner–Munzel probability versus relative chromosome-arm position and found negative correlations that were significant for most chromosome arms (Fig. 2a,b and Extended Data Fig. 2b,c). This suggests a model in which Cas9 editing can cause chromosomal truncations resulting in mixed phenotypes due to multiple gene deletions (Fig. 2c).

We sought to test this hypothesis by searching for and quantifying chromosome-arm truncations in sequencing data, paralleling similar searches for truncations and deletions in literature[9,10]. We reanalyzed two scRNA-seq datasets uniformly reprocessed as part of the scPerturb resource[34,35], which profiled the effects of CRISPR–Cas9 gene knockout in the THP-1 leukemia line and in melanoma-derived melanocytes, respectively[36,37]. Following previous work[9,10], we assessed deletions in this data by identifying the genes, which, when targeted by CRISPR–Cas9, result in substantial, significant copy number loss in more than 70% of the 150 genes near the cut site in the 3′ or 5′ direction (Methods). Across these datasets, editing at 4–25% of targeted genes resulted in copy number loss. Moreover, those losses were significantly more likely to occur in the telomeric direction (Fisher's exact test $P = 1.4 \times 10^{-5}$), further supporting the model of chromosome-arm truncations and supporting the cell-type independence of proximity bias (Table 1). As reported in previous work[9], for targets with a called loss, only a fraction of cells exhibited a deletion (mean 4.3% and maximum 15.1%) (Supplementary Table 2). Figure 2d,e highlights the enrichment for loss in the telomeric direction for each of the two

datasets examined, by showing whole-genome copy number calls for the cells that exhibited a deletion.

To further validate the finding of chromosomal arm truncations, we searched an internal Recursion database of HUVEC bulk RNA sequencing data and focused on a high-replicate set of 45 targeted genes, each treated with a single intron-targeting guide in 63 replicate samples compared with a no-guide reference pool of 3,320 samples (average 1.3 million unique reads per well). Comparing copy number calls from Cas9-edited wells to Cas9-free controls, we observed multiple loci enriched for deletions between the target cut site and the telomere, including the genes *ZNF394* (located on chromosome 7q) and *RCAN3* (on chromosome 1p) (Fig. 2f, Extended Data Fig. 2d and Supplementary Table 1). Although the number of loci found in this search was limited, these events are expected to be rare and, therefore, difficult to detect in bulk assays, particularly by RNA rather than DNA sequencing.

## Proximity bias confounds therapeutic target identification

A key application of genome-wide knockout screening is in mapping of biological pathways, particularly for therapeutic target discovery. Consequently, we investigated the potential impact of proximity bias on target discovery in a widely used, publicly available resource.

Project Achilles has performed genome-wide CRISPR screens of cell survival in hundreds of cancer cell lines in an effort to identify potentially druggable essential genes for a range of tumor types, contributing to the DepMap[11]. We surmised that if DepMap CRISPR screens were also affected by proximity bias, then it would manifest as patterns of essentiality across cell types that cluster unexpectedly by genomic proximity. To that end, we built genome-wide maps from DepMap CRISPR 19Q3 data; in these maps, each gene was characterized by a vector representing its essentiality in each of the 625 tested cell lines rather than as a vector of morphological features (Fig. 3a). Visual

examination and quantification of the DepMap CRISPR map confirms the presence of arm-scale proximity bias (Fig. 3b,c), and the proximity bias effect is maintained in a newer version of these data (22Q4), which controls for CNVs[17].

While correlations between gene dependency and genomic location have been reported before and several correction methods have been implemented[14–17,38], the effect was thought to be primarily driven by copy number variation in these cancer cell lines. Since we observe similar effects in copy number-normal cell lines, we sought to disentangle CNV-based effects from the proximity bias effect using the following procedure (Methods). Beginning with the full set of 1,078 cell lines in the CRISPR–Cas9 DepMap 22Q4 data, we first looked for subsets of cell lines that were free from CNVs on each autosomal chromosome arm. Then, for each pair of those arms, we intersected the cell line sets and assessed proximity bias by computing the Brunner–Munzel probability of within-arm cosine similarities exceeding between-arm cosine similarities (741 arm pairs; intersection cell line counts minimum, maximum and mean of 73, 314 and 174, respectively) (Supplementary Tables 5 and 6). These values were compared with probabilities from the same process but using all cell lines in both the CRISPR–Cas9 and short hairpin RNA (shRNA) data (190 cell lines) (Extended Data Fig. 3b). Controlling for copy number in this way significantly reduces proximity bias (Mann–Whitney $U$, $P$ value $<1 \times 10^{-10}$) but fails to eliminate it, with all arm pairs showing arm-level Brunner–Munzel probabilities above 0.5. Since restricting to fewer cell lines reduces the power to detect gene–gene interactions in general, we also see a reduction in Brunner–Munzel probabilities toward 0.5 when subsampling cell lines randomly. While a further reduction is observed in the cell lines with very few CNVs, there is still significantly more proximity bias in CNV-controlled CRISPR–Cas9 arm pairs than in pairs formed from shRNA data in which CNVs are still present, but there is no DNA cutting, suggesting that CNVs alone cannot explain the effects observed (Mann–Whitney $U$, $P$ value $<1 \times 10^{-10}$) (Extended Data Fig. 3b). Finally, we note that the DepMap Chronos pipeline for CRISPR data excludes guides which map to multiple regions, so these results cannot be explained by multitargeting guides[17].

In addition, we examined the results from shinyDepMap[39], which sought to cluster genes with similar dependencies to identify druggable targets and pathways using the 19Q3 data. Examination of 16,941 inferred gene–gene relationships from shinyDepMap CRISPR data revealed that a large number of putative relationships inferred are within chromosomal arms. Comparing the odds of identifying intra- versus inter-arm connectivity in the shinyDepMap clusters with other databases of known biological relationships derived from pathway or protein complex data revealed that shinyDepMap contains far more intra-arm annotated relationships than other sources (Fisher exact test odds ratio of 0.068, $P < 0.0001$). This suggests that results of downstream DepMap CRISPR analyses may be significantly confounded by proximity bias (Extended Data Fig. 3c,d).

Finally, we sought to identify cancer-specific false-positive dependency calls due to proximity bias and not CNVs. If the hypothesis that proximity bias is caused by telomeric truncations were correct, some unexpressed genes centromeric of driver genes would spuriously appear as essential, since occasional truncations telomeric of the targeted unexpressed gene would also delete the true driver. To explore this, we used the DepMap 22Q4 dependency data and first stratified cell lines by their annotated cancer subtype; then, for each gene, we restricted the cell lines to those without CNVs (copy number within (1.75, 2.25)) and tested for differences in dependency for that subtype versus all other cell lines. Examining three cancer subtypes, we found a number of genes centromeric of known subtype-specific driver genes that have low expression (transcripts per million reads <0.3) but nevertheless exhibit significantly higher dependency (that is, appear more essential) than in other subtypes (Benjamini–Hochberg adjusted $t$-test, $P < 0.01$). For example, on chr2p for renal cell carcinoma, four

genes centromeric of the driver *EPAS1* (ref. 40) (*C2orf73*, *ARHGAP25*, *VAX2* and *LRRTM4*) satisfy these criteria, as do two genes on chr18q for B-lymphoblastic leukemia and lymphoma centromeric to the driver *BCL2* (ref. 41) (*ELOA2* and *GRP*) and three genes on chr2p for neuroblastoma genes centromeric to the driver *SOX11* (ref. 42) (*KCNF1*, *NTSR2* and *FAM166C*) (Supplementary Table 7). This suggests that these essentiality annotations may be spurious and actually are caused by proximity bias-related truncations of nearby true driver genes.

## Proximity bias is dependent on Cas9 nuclease activity

Given the proposed model of large truncations, we sought to confirm whether proximity bias is dependent on nuclease activity of Cas9 by analyzing CRISPR interference (CRISPRi)[43] and shRNA screens[44]. We extended our analysis of scRNA-seq CRISPR–Cas9 datasets from scPerturb[35] to three CRISPRi datasets[45–47] and found that in contrast to CRISPR–Cas9-perturbed cells, in which 4.2–25% of genes resulted in large chromosomal losses when targeted (Table 1), only up to 2.6% of target genes were observed to have such losses across three CRISPRi datasets (Table 1 and Supplementary Table 2). We also examined whether there is evidence of telomeric loss enrichment in the significantly smaller proportion of genes showing loss in the CRISPRi datasets, and our findings were negative (Fisher's exact test, $P = 0.68$). Additionally, a map built from DepMap shRNA screening data did not show substantial proximity bias, suggesting that the effect arises as a specific consequence of CRISPR–Cas9 editing (Extended Data Fig. 4a,b).

## DepMap data connects proximity bias and cell cycle

As the DepMap data profiled a wide range of cell lines with diverse genetic backgrounds, we hypothesized that by stratifying cell lines according to genetic features and constructing maps out of slices of these data, we may be able to elucidate the potential biological mechanisms behind, and the mediators of, proximity bias.

*TP53* expression has been suggested as a marker for reduced aneuploidy in CRISPR–Cas9 editing[9,31,48], and previous work has established that p53 activity reduces CRISPR–Cas9 editing efficiency through activation of DNA repair and apoptosis[49–53]. Thus, loss of *TP53* would be expected to increase proximity bias by increasing the rate of chromosomal arm truncations. We stratified DepMap cell lines by *TP53* loss-of-function (LOF) status and found significantly increased proximity bias ($t$-test, $P <1 \times 10^{-10}$) in a CRISPR map built from putatively *TP53*-null cell lines (LOF) compared with one built using only cell lines with putatively functional wild-type (WT) *TP53* (Fig. 4a).

Next, we searched for additional gene mediators while controlling for *TP53* status across eight splits: for genes in which putative LOF or amplification (AMP) either increased or decreased proximity bias, in either a *TP53* null or functional background (Supplementary Table 3). Several genes showed interesting behaviors that support their known functions in cell cycle and *TP53* regulation (Fig. 4b,c and Supplementary Table 3). In both the *TP53* WT and *TP53* LOF settings, we found that loss of *CDKN2A* or *CDKN2B* significantly increases proximity bias while *CDKN2C* AMP decreases proximity bias in a *TP53* LOF background (cell line bootstrap $t$-test Bonferroni-corrected $P < 0.05$; DepMap did not have sufficient cell lines for us to test *CDKN2C* in the *TP53* WT setting). This suggests that these cell cycle regulators[54–56] act independently of p53. Conversely, AMPs of the *TP53* regulators *MDM2* and *MDM4* (ref. 57) show differential effects on proximity bias depending on the *TP53* background. Both AMPs increase proximity bias when a functional *TP53* is present, but *MDM2* AMP has no effect in the *TP53* LOF setting, while *MDM4* AMP decreases proximity bias in that environment. This suggests that the effect of *MDM2* on proximity bias is entirely mediated through *TP53*.

Additionally, we found that because of large-scale CNVs, identifying drivers of proximity bias is itself affected by chromosome-position effects, making it difficult to confidently fine-map individual driver genes within a genomic region. For example, *BTG2* surfaced as a

**a**

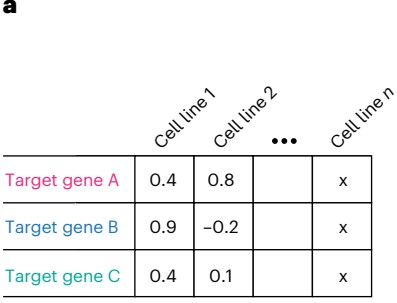

**b**

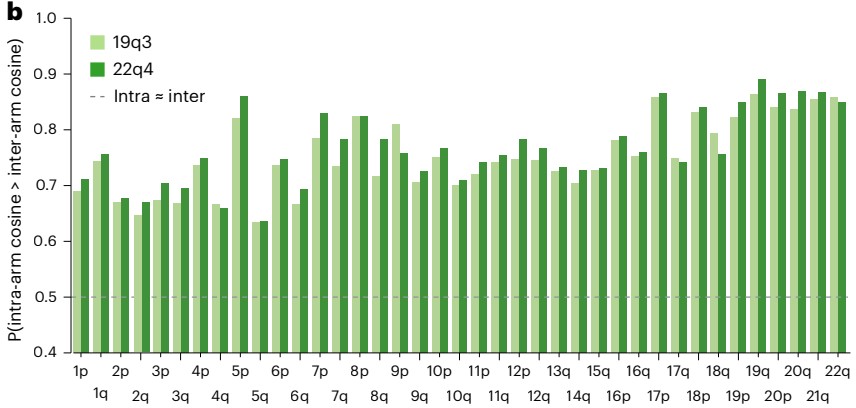

**c**

**Fig. 3 | CRISPR–Cas9 screens of cancer gene essentiality are significantly confounded by proximity bias. a**, A schematic of the table of feature data used in the DepMap analysis. Each row corresponds to a target gene and each column to a cell line. The values in the table are dependency measures of the survival sensitivity for the cell line when the target gene is knocked out with CRISPR–Cas9. We assessed similarity between targets by calculating the cosine similarity between rows. **b**, A bar plot of proximity bias metrics (one-sided Brunner–Munzel probabilities) for each chromosome arm for the DepMap 19Q3 and 22Q4 datasets. The values above 0.5 indicate elevated intra-arm similarity, and all chromosome arms are significant with Bonferroni correction (*P* < 0.001). **c**, Pairwise cosine similarity between DepMap targets (19Q3 above the diagonal and 22Q4 below the diagonal) ordered by chromosome position across the human genome and quantile normalized to a normal distribution with mean 0 and standard deviation 0.2. The solid lines represent chromosome boundaries and the dashed lines indicate centromeres.

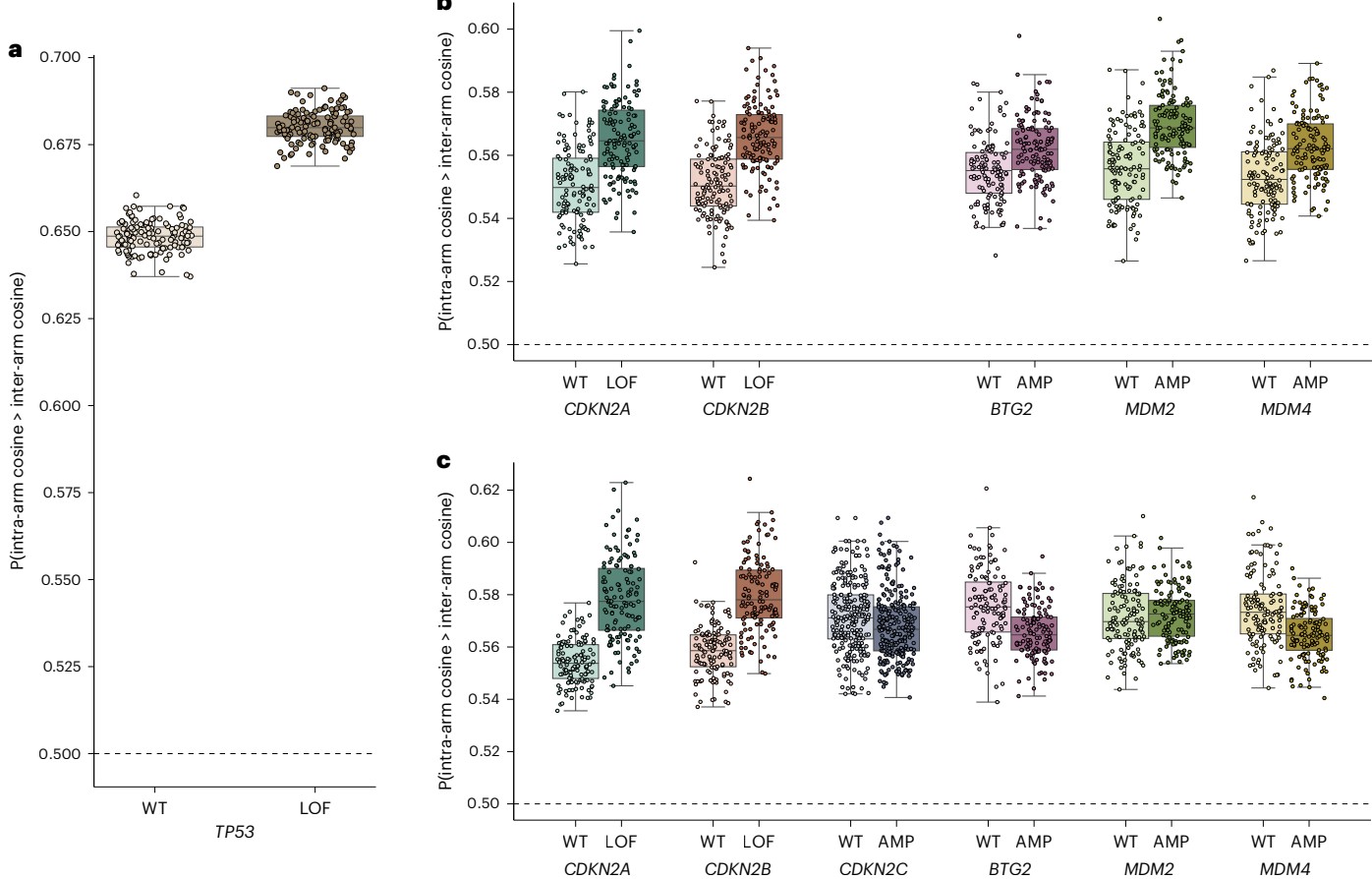

**Fig. 4 | Role of *TP53*, cell-cycle and replication-associated genes in proximity bias.** Box and scatter plots of whole-genome-level proximity bias quantification by one-sided Brunner–Munzel intra-arm versus inter-arm probability from DepMap 22Q4 data with the cell lines stratified by gene status. Each point represents a bootstrap sample of cell lines; 128 bootstraps were run for each condition. The box plots show the median and lower and upper quartile with whiskers extending to the furthest points within 1.5 times the interquartile range. The *P* values for two-sided *t*-tests with Bonferroni correction applied are given, and the detailed test statistics are given in Supplementary Table 3. Horizontal dashed lines at 0.5 indicate the expected baseline if no proximity bias effect was present. **a**, Comparison between cell lines with WT *TP53* and those with *TP53* LOF ($n = 266$ and $277$ cell lines, respectively, 212 sampled in each bootstrap; $P < 1 \times 10^{-10}$). **b**, Selected genes and conditions LOF or AMP in *TP53* WT background (*CDKN2A*: WT $n = 112$, LOF $n = 136$, $P < 1 \times 10^{-10}$; *CDKN2B*: WT $n = 124$, LOF $n = 132$, $P < 1 \times 10^{-10}$; *BTG2*: WT $n = 173$, AMP $n = 87$, $P = 1 \times 10^{-6}$; *MDM2*: WT $n = 210$, AMP $n = 52$, $P < 1 \times 10^{-10}$; *MDM4*: WT $n = 174$, AMP $n = 87$, $P = 2 \times 10^{-10}$; $n = 20$ cell lines were sampled in each bootstrap for all genes). **c**, Selected genes and conditions in *TP53* LOF background (*CDKN2A*: WT $n = 82$, LOF $n = 156$, $P < 1 \times 10^{-10}$; *CDKN2B*: WT $n = 103$, LOF $n = 154$, $P < 1 \times 10^{-10}$; *BTG2*: WT $n = 188$, AMP $n = 81$, $P = 5 \times 10^{-10}$; *MDM2*: WT $n = 218$, AMP $n = 43$, $P = 1$; *MDM4*: WT $n = 187$, AMP $n = 79$, $P = 6 \times 10^{-9}$; $n = 20$ cell lines were sampled in each bootstrap for all genes). The genes with fewer than 25 cell lines in a given condition are not shown.

potential driver but is located only 1.2 Mb in the centromeric direction from *MDM4* on chromosome 1 and appears to mimic its impact on proximity bias in both the *TP53* WT and LOF conditions. Upon closer inspection, we find that all 15 of the genes between these two, with sufficient data to assess, show the same pattern despite no known cancer or *TP53* associations (Extended Data Fig. 5a,b).

We also looked for enriched biological processes among the genes with largest impacts on proximity bias in each of the above contexts using ShinyGO (v0.77)[58]. Selecting genes with mean differences in Brunner–Munzel probabilities between WT and LOF and AMP conditions of less than −0.1 or greater than 0.2, we found the strongest associations in the *TP53* WT setting were with 'regulation of cell population proliferation' and 'positive regulation of cell population proliferation', where AMP of 34 and 25 genes, respectively, show increased proximity bias ($P < 1 \times 10^{-10}$). In the *TP53* LOF setting, the strongest associations were with 'regulation of apoptotic processes' and, again, 'regulation of cell population proliferation' ($P < 1 \times 10^{-10}$) where, in both cases, AMP for 28 genes shows increased proximity bias (Supplementary Table 4). This supports the hypothesis that proximity bias is driven by chromosome-arm truncations and suggests a mechanism in which

inhibition of apoptosis may lead to unrepaired double-strand breaks and loss of acentric chromosome-arm fragments during mitosis[9,10].

### Geometric correction reduces proximity bias

Given that the proximity bias effect appears to be largely localized within chromosome arms, we hypothesized that applying a chromosome-arm correction to rxrx3 and cpg0016 maps might mitigate the unwanted signal. To that end, we adjusted the vector representation for each gene by subtracting an estimated representation of the chromosome arm in which the gene is located built using unexpressed genes (Methods). This significantly reduces the proximity bias effect, both globally and per chromosome arm (Fig. 5a–d) while maintaining or improving genome-wide benchmarking metrics in both datasets (Fig. 5e). Interestingly, the recall of annotated within-arm relationships decreases with the chromosome-arm correction (Fig. 5e), but this is outweighed by improved recall on the larger number of between-arm annotated relationships, suggesting that the proximity bias effect can confound such benchmarking efforts if it is not taken into account.

Following the preprint publication of this work, DepMap released the 23Q2 revision of its Project Achilles CRISPR screens, incorporating

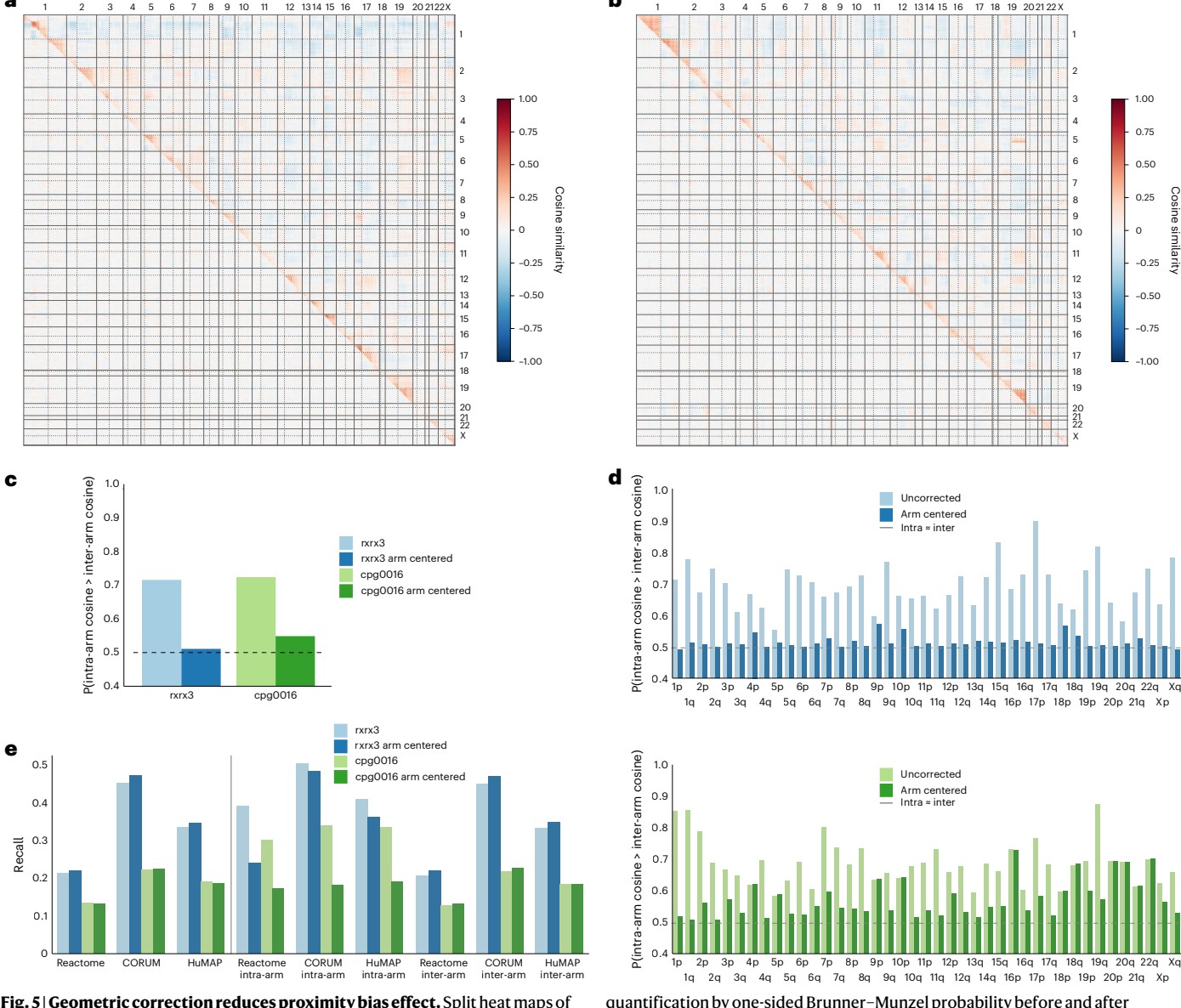

**Fig. 5 | Geometric correction reduces proximity bias effect.** Split heat maps of pairwise cosine similarities before (above diagonal) and after (below diagonal) chromosome-arm correction. **a**, The split heat map for rxrx3. **b**, The split heat map for cpg0016. **c**, A bar plot of genome-wide proximity bias quantification by Brunner–Munzel test statistic before and after chromosome-arm correction for rxrx3 and cpg0016. The correction greatly reduces the probability of within-arm relationships having a stronger cosine similarity than between-arm relationships. The horizontal dashed line at 0.5 indicates the expected baseline if no proximity bias effect was present. **d**, A bar plot of chromosome arm-level proximity bias

quantification by one-sided Brunner–Munzel probability before and after chromosome-arm correction for rxrx3 (top) and cpg0016 (bottom). The correction reduces the probability of within-arm relationships having a stronger cosine similarity than between-arm relationships for most chromosome arms. Horizontal dashed lines at 0.5 indicate the expected baseline if no proximity bias effect was present. **e**, Annotated relationship recall benchmarking metrics with and without chromosome-arm correction, both at the global scale (left of vertical line) and stratified by within-arm or between-arm relationships.

a correction similar to that suggested in this section (https://forum.depmap.org/t/announcing-the-23q2-release/2518). A genome-wide map built from this data similarly reduces proximity bias, demonstrating the generality of this geometric correction across modalities (Extended Data Fig. 3a). Additionally, the potential false positive driver genes for specific cancer subtypes discussed in the previous section are reduced with six of the nine highlighted genes no longer showing a subtype-specific dependence (Supplementary Tables 7 and 8).

## Discussion

Since its discovery, the CRISPR–Cas9 editing system has become a valuable research tool; however, deep characterization has revealed potential issues arising from undesired on-target effects. In this work,

we use cellular phenomics to systematically profile CRISPR-induced gene knockouts for virtually all human protein-coding genes in a primary human cell type and have replicated our findings across cell types, assay contexts and molecular follow-up. We discover an undesired on-target effect driven by a small fraction of cells that results in knockout phenotypes displaying a 'proximity bias' that probably arises from chromosomal truncations and is ubiquitous across cell types, genetic loci and measurement modalities but can be computationally corrected given proper controls.

This refines prior work that asserted aneuploidy and chromosome truncation as a potential consequence of CRISPR–Cas9 editing in T cells[9,10], primarily by focusing on the *TRAC* locus (14q11.2, near the centromere of the acrocentric chromosome 14). Reanalyzing Perturb

Cellular Indexing of Transcriptomes and Epitopes (Perturb-CITE) sequencing data from melanoma cells[37], we found evidence for similar occasional loss of the entire chromosome 21 arising from editing of *SLC19A1*, located on the q arm (21q22.3) (Fig. 2e and Supplementary Table 2). In our transcriptomic analysis, chromosomal truncations were primarily seen to proceed in the direction away from the centromere, but since it is well established in medical genetics that the short arms of acrocentric chromosomes 13, 14, 15, 21 and 22 are nonessential[59], this suggests that the observation of whole-chromosome loss in T cells is probably a specific artifact of editing pericentromeric loci on acrocentric chromosomes.

The apparent generality of undesired on-target effects from CRISPR–Cas9 editing raises potential concerns for both functional genetic screening and therapeutic gene editing. Prior literature has primarily examined cell lines[31–33] or zygotes, embryos or embryonic stem cells[4,6–8,30], which have varying DNA damage responses, so continuing to establish the importance of these effects in somatic primary cells will be important[49,50,52]. More recent work[9,10] has shown recurrent aneuploidy in ex vivo edited human T cells and suggested that protocols inducing *TP53* expression before editing may be protective for chromosome truncation. However, *TP53* induction may not be feasible in many settings. In particular, somatic loss of *TP53* has been observed to increase in frequency with age in a variety of nonmalignant tissues, including colonic epithelium and blood[60–62], suggesting that potential risks related to in vivo CRISPR–Cas9 editing may be age dependent. Although no negative consequences due to unintended effects of CRISPR cutting have yet been documented in patients, further research is necessary to detect and quantify the presence of chromosomal losses in in vivo editing to maximize patient safety.

Our chromosome-arm truncation hypothesis is consistent with recent findings from other groups[7,32], based on transcriptomic evidence and DepMap[11] analysis, and suggests a mechanism involving CRISPR–Cas9-induced losses in a subpopulation of cells, with increased mitosis potentially amplifying the effect. We find higher rates of deletions in CRISPR–Cas9 RNA sequencing data relative to shRNA on both sides of the cut but are more common in the telomeric direction. While previous work analyzing dependency studies in cancer cell lines found similar effects due to copy number variations[14–16], we demonstrate that this effect is largely independent of copy number by quantifying its presence in primary cell types and in regions of cancer cell lines that lack CNVs. The mechanism proposed here generates testable hypotheses for future research, exploring the impact of mitogens, cell cycle inhibitors or repeated passaging on deletion rates and suggests that highly mitotic cell types may experience more proximity bias in CRISPR–Cas9 functional genomics screens than slowly or nondividing cells.

Additionally, inspection of whole-genome similarity maps similar to Fig. 1d–f suggests that proximity bias patterns are more complex than just increased similarity within chromosome arms and that these patterns probably differ between cell types. This may be due to a wide variety of factors including differences in susceptibility to truncation, epigenetic state influencing Cas9 efficiency, gene haploinsufficiency, gene essentiality and the strength of phenotypic effects caused by genes telomeric from the target loci. To deconvolve these effects, a further investigation across many cell types with consistent data collection and processing is needed.

Finally, we suggest a correction strategy that estimates and removes the confounding signal on each chromosome arm using unexpressed genes. This highlights the advantages of taking a genome-wide view and suggests control strategies both for large gene surveys and for more targeted screening. Beyond this geometric correction, a wide range of other mitigation strategies may be developed to combat proximity bias. From a biological or biochemical perspective, it is probable that the use of noncutting perturbations—for example, CRISPRi[43], CRISPRoff[63], base editors or RNA-targeting perturbations such as Cas13d[64]—would circumvent proximity bias; however, some recent studies suggest that base and prime editors can induce double-strand breaks and associated deletions or translocations[65]. With cutting-based CRISPR assays, activation of p53 or DNA repair pathways (for example, through nutlin pretreatment)[66] may mitigate this effect, as may the addition of free nucleotides or optimizing the timing of experimental steps[67], modifying Cas9 constructs[68] or extending the 5′ end of single guide RNAs with cytosine bases[69]. Additionally, given that these effects are probably driven by a relatively small subpopulation of cells, improved data cleaning strategies may also prove fruitful. Ideas here include the filtering of subsets of cells in transcriptomics or patches of images in phenomics or utilizing loss functions during neural network training to ignore populations of affected cells. While each method has particular limitations (for example, durability, specificity and computational intensity), the quantification methods presented in this study can be used to judge effectiveness and to drive innovation in this area.

## Online content

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

## Methods

This research complies with all relevant ethical regulations as approved by Recursion Pharmaceuticals.

### Cell culture

HUVEC umbilical vein endothelial cells (Lonza, C2519A) at early passage are expanded within an acceptable window of in vitro culture in single-use bioreactor systems that provide 250,000 $cm^2$ of growth surface. This results in a yield of $10 \times 10^9$ cells to screen up to 4,000 1,536-well plates. HUVEC are produced and banked in vapor-phase liquid nitrogen and successfully seeded into high-throughput screens directly from stasis post editing. HUVEC are seeded into 1,536-well microplates (Greiner, 789866) via Multidrop (Thermo Fisher) and incubated at 37 °C in 5% $CO_2$ for the duration of the experiment.

### CRISPR–Cas9 editing

Custom-designed Alt-R CRISPR–Cas9 reagents were purchased from Integrated DNA Technologies and prepared following the manufacturer's guidelines and protocols (Alt-R CRISPR–Cas9 crRNA, Alt-R CRISPR–Cas9 trans-activating RNA (tracrRNA) cat. no. 1072534, Alt-R S.p. Cas9 Nuclease V3, cat. no. 1081059). Alt-R CRISPR–Cas9 crRNA was duplexed to Alt-R CRISPR–Cas9 tracrRNA and then combined with Alt-R S.p. Cas9 Nuclease V3, following Integrated DNA Technologies guidelines, to form a functional CRISPR–RNP (ribonucleoprotein) complex. This CRISPR–RNP complex was transfected into cells with a proprietary lipofection-based process for high-throughput application.

To control for and filter nonproximal off-target effects of individual guides, each gene was targeted with 4–12 nonoverlapping guides (89% of genes targeted by six guides), for a total of 101,029 guides. Each guide was assessed independently in an arrayed format, typically with 24 total replicate wells per guide across two executional batches.

### Phenomic imaging

The plates were stained using a modified cell painting protocol[19]. The cells were treated with MitoTracker deep red (Thermo, M22426) for 35 m; fixed in 3–5% paraformaldehyde; permeabilized with 0.25% Triton X100; stained with Hoechst 33342 (Thermo), Alexa Fluor 568 Phalloidin (Thermo), Alexa Fluor 555 wheat germ agglutinin (Thermo), Alexa Fluor 488 concanavalin A (Thermo) and SYTO 14 (Thermo) for 35 min at room temperature; then washed and stored in Hanks' balanced salt solution + 0.02% sodium azide. The images were acquired with ImageXpress micro confocal microscopes (Molecular Devices) in wide field mode using a PlanApo 10× 0.45 numerical aperture objective and Spectra-3 light-emitting diode light engine (Lumencor). For the sake of acquisition speed, six-channel imaging was accomplished using three combinations of two dichroic mirrors and three emission filters.

### Phenomic analysis

All images were uploaded to cloud storage and featurized by embedding them with a proprietary convolutional neural network trained on the public RxRx1 dataset using Google Cloud Platform as described in a previous work[70]. The images are captured at 2,048 × 2,048 pixels and divided into 16 tiled patches, which are each embedded separately. Those embeddings are averaged to create a single representation for each imaged well.

### Generation of gene-level representations for rxrx3 HUVEC data

For this screen, Recursion ran 176 12-plate experiments in 1,536-well plates, generating 24 images per guide for a total of 101,029 guides and 17,065 genes. The embedding vectors for each image were centered on a set of perturbation controls, aligned using typical variance normalization and aggregated to the gene level as described in Celik et al.[20]. The externally released version of this data described in Fay et al.[22] contains all the same gene guides but was processed with an older pipeline and contains fewer replicates per guide (18), so there may be small discrepancies with the data shown here.

### Generation of gene-level representations for cpg0016 U2OS data

The well-level aggregated CellProfiler profiles were downloaded from the Cell Painting Gallery[23]. The 'Image' CellProfiler and 'ObjectNumber' features were discarded, and the remaining features were normalized by plate. A principal component analysis was performed using a 98% variance cutoff to reduce the dimensionality of the data, followed by an additional plate normalization step. The experimental replicates were aggregated by taking the mean to yield a feature representation per gene.

### Normalization of cosine distributions across maps

To make different heatmaps (for example, rxrx3 HUVEC data and cpg0016 U2OS data) visually comparable, cosine similarity values for each map were quantile normalized to a normal distribution with mean zero and standard deviation 0.2 for display purposes only.

### Benchmarking of known relationships

To assess how well a map embedding recapitulates known biology, we calculated recall measures on known pairwise relationships from annotated sources (Reactome[25], HuMap[26] and CORUM[27]) as follows. Given pairwise cosine similarities between the aggregated perturbation embeddings of all perturbed genes, we selected the top 5% and bottom 5% of gene pairs (excluding self-relationships) from the cosine similarity distribution as 'predicted relationships'. We then calculated the recall as the proportion of these predicted relationships over all relationships in the annotation source. If annotated relationships were spread randomly throughout the cosine distribution, this would produce a recall of 0.1, so that value is used as a baseline. For Fig. 1c, we have 314, 460 and 530 annotated relationships within chromosome arms and 6,713, 11,502 and 11,376 between chromosome arms for Reactome, HuMap and CORUM, respectively.

### Statistics and reproducibility

No statistical method was used to predetermine sample size. No data were excluded from the analyses. The experiments were not randomized and investigators were not blinded to allocation during experiments and outcome assessment.

Analysis was performed using Python (v3.9), numpy (v1.22.3), pandas (v1.4.2), scipy (v1.10.1) and statsmodels (v0.13.2). The visualizations were generated using matplotlib (v3.5.2), scikit-image (v0.18.3) and seaborn (v0.12.3). scRNA-seq data were processed with STAR (v2.7.7a) and scanpy (v1.9.3), and the CNV elements were determined with infercnvpy (v0.4.1). The list of cancer genes was downloaded from OncoKB (v4.4), https://www.oncokb.org/cancer-genes.

To quantify the level of proximity bias for cell-line splits in the DepMap data, we computed the effect size of intra-arm relationships being larger than inter-arm relationships, according to a Monte Carlo variant of the Brunner–Munzel test[29]. In particular, we compute:

$$P(\text{intra arm} > \text{inter arm}) = \frac{\sum_{i=1}^{N} \text{rank}(\text{intra arm}_i)}{NM} - \frac{N+1}{2M}$$

where $N$ is the number of intra-arm samples, $M$ is the number of inter-arm samples and $\text{rank}(x)$ is the index of sample $x$ when all samples are sorted, with ties being assigned their average rank. To perform bootstrapping, we set $N = M$ and repeatedly sampled $N$ random pairs of genes from both the intra-arm and inter-arm populations for $T$ trials. The dependency scores for each gene pair across cell lines are used to compute a cosine similarity, which is then used as the ranking metric. Except where otherwise noted, we utilized $N = 500$ and $T = 100$. For the final score, we took the empirical mean over these trials. The numbers

of cell lines used in each split and other statistical details are included in Supplementary Table 3.

To quantify proximity bias at the genome level, the Brunner–Munzel test statistic was computed between the full inter-arm and intra-arm cosine similarity distributions across all chromosome arms (without sampling). The statistic estimates the probability that the intra-arm cosine similarity is greater than the inter-arm cosine similarity, and the P values shown in Fig. 1g,h and Extended Data Fig. 4b are one tailed and Bonferroni corrected. For arm-level metrics, we restricted the distributions to gene pairs within a given arm versus pairs with one gene on that arm. The sample sizes can be determined by the number of genes on each chromosome arm, which are given in Supplementary Table 5 for the rxrx3, cpg0016 and DepMap 22Q4 data, and tests were performed only for chromosome arms with at least 20 within-chromosome-arm pairs.

Gene-level proximity bias quantification was computed by the Brunner–Munzel test between all cosine similarities of the gene to all other genes on the same arm and the cosine similarities to genes on other arms (no sampling). The correlation between proximity bias and rank gene location between the telomere and the centromere was computed using the Spearman rank correlation of the Brunner–Munzel statistics and the ordered position of the genes on the chromosome arm. For Fig. 2b (rxrx3 data) and Extended Data Fig. 2c (cpg0016 data), the sample sizes are given in Supplementary Table 6.

### Analysis of public scRNA-seq data

Files containing scRNA-seq AnnData objects for two CRISPR–Cas9 and three CRISPRi screens were downloaded from PapalexiSatija2021_eccite_arrayed_RNA.h5ad[36], FrangiehIzar2021_RNA.h5ad[37], ReplogleWeissman2022_rpe1.h5ad[44], TianKampmann2021_CRISPRi.h5ad[45] and AdamsonWeissman2016_GSM2406681_10×010.h5ad[46] as harmonized by the scPerturb study[35]. Each dataset was loaded using the 'scanpy' package[71] (v1.9.3). and we determined the CNV events using the 'infercnvpy' package (v0.4.1), which is a scalable implementation of 'inferCNV' of the Trinity CTAT project (https://github.com/NCIP/Trinity_CTAT). For each of the datasets, we identified genes that, when perturbed, led to chromosomal loss proximal to the target gene (Table 1 and Supplementary Table 2). A perturbed gene is identified as resulting in proximal chromosomal loss in a cell if 70% or more of the neighboring 150 genes in the same chromosome are lost (that is, inferred CNV value of ≤−0.05) in that cell.

It is crucial to ensure that the chromosomal loss at a targeted locus is specifically due to the perturbation at that locus and is not a nonspecific loss commonly observed when other genes on distal chromosomes are targeted. Observed proximal loss when a gene G is perturbed is considered specific if the fraction of cells exhibiting loss near G is a minimum of three standard deviations away from the average fraction of cells demonstrating chromosomal loss near G when any gene within the dataset is perturbed. For each of the genes called using this process, the fraction of impacted cells (that is, cells that lose more than 70% of the 150 genes near the perturbed gene) is reported (Supplementary Table 2). Finally, we generate the heat maps in Fig. 2d,e using the 'infercnvpy' package (https://github.com/icbi-lab/infercnvpy).

### Analysis of bulk RNA sequencing data

Illumina reads were aligned to the hg38 reference and gene-level counts generated for each sample using the gencode_v33 gene annotation set and stored in AnnData objects together with sample perturbation metadata using STAR v2.7.7a (ref. 72) and scanpy (v1.9.3)[71]. The CNV events and determination of chromosomal loss near the on-target cut site were determined using the method described above for the scRNA-seq analysis. A total of 45 intron-cutting CRISPR guide perturbations were tested in HUVEC cells using this method (Supplementary Table 1).

### Analysis of DepMap data

Four datasets from DepMap (https://depmap.org/portal) were analyzed: CRISPR–Cas9 23Q2 (Chronos pipeline with arm normalization correction), CRISPR–Cas9 22Q4 (Chronos pipeline), CRISPR–Cas9 19Q3 (CERES pipeline) and shRNA (DEMETER2 pipeline)[17,18]. For each gene, we treated the dependency scores across different cell lines as a feature vector and computed the cosine similarity to other genes in the dataset (Fig. 3a). To reduce the bias toward essential genes from cosine similarity computation, we recentered the dependency scores for each gene by subtracting the mean from all cell lines. The cosine similarity values were then quantile normalized to a normal distribution with mean zero and standard deviation 0.2 for display purposes only.

To disentangle proximity-bias driven effects from CNVs dependencies, we reanalyzed the DepMap 22Q4 data (which has been corrected for copy number using the Chronos pipeline[17]) by estimating the Brunner–Munzel probabilities across pairs of chromosome arms with almost no CNVs. For each pair of autosomal chromosome arms, we restricted to cell lines where less than 1% of genes had copy-number calls outside of (1.75, 2.25) (Supplementary Tables 5 and 6) and calculated the arm-level Brunner–Munzel probabilities (without sampling) for each pair (1,482 total values). These were then compared with Brunner–Munzel probabilities using all cell lines in both the full CRISPR–Cas9 data and shRNA data, as well as to randomly sampled cell lines matching the number of cell lines without CNVs (ten random sampling runs) (Extended Data Fig. 3b).

Additionally, we performed an analysis of the difference in proximity bias effect observed in WT cell lines as compared with AMP or LOF cell lines using the 22Q4 DepMap data. This was further stratified by looking at both a TP53 WT background and a TP53 partial LOF background. This was performed by restricting to the cell lines that are TP53 WT or partial LOF (copy number ≤1.5) before computing the proximity bias score. To select the cell lines matching LOF or AMP, we first subset to cell lines that have copy number ≤1.5 or ≥2.5 for LOF and AMP, respectively. Then, for AMP, we additionally subset to cell lines that do not have a nonsense or frame shift mutation, as these cell lines may have LOF despite the AMP. To control for different numbers of cell lines in different conditions, we computed a bootstrap version of the Brunner–Munzel proximity bias metric described above with an additional level of sampling. This consists of taking a random sample of 20 of cell lines in each condition, constructing maps and calculating the Brunner–Munzel probabilities for $S = 4$ trials. Additionally, we increase the number of trials $T$ (of genes used to compute cosine similarity) to 200 and exclude any conditions with fewer than 25 cell lines. Once this metric was computed for all 602 genes with a sufficient number of cell lines to meet the above conditions, we computed the difference between the WT and each mutant condition in each background condition, subset to the top 200 genes and repeat the computation with $S = 32$ trials (Supplementary Table 3). The initial list of cancer genes was downloaded from OncoKB[73] (v4.4, https://www.oncokb.org/cancer-genes).

Gene-set enrichment was conducted for Gene Oncology biological process with ShinyGO (v0.77)[58] using all genes with mean differences in whole-genome level Brunner–Munzel statistics above 0.2 for increases in proximity bias and below −0.1 for decreases in proximity bias in each condition. All default settings were used (false discovery rate cutoff of 0.05; number of processes to show; process size: minimum 2, maximum 2,000; selected by false discovery rate and sorted by fold enrichment) (Supplementary Table 4).

For the data in Supplementary Tables 7 and 8, the cell lines were grouped by disease annotations and then for each gene, we performed t-tests on the dependency values, first between all cell lines with that annotation and then after restricting to cell lines with copy number calls within (1.75, 2.25). False discovery rate correction was applied to all tests (Benjamini–Hochberg). We report all significant genes in B-lymphoblastic leukemia and lymphoma, neuroblastoma and renal

cell carcinoma for both the 22Q4 and 23Q2 datasets along with the gene expression values (in transcripts per million reads). The 23Q2 data have a correction applied for the proximity bias effect, so differences between these two tables highlight the reduction in potential false-positive disease-specific driver genes.

## Geometric method for proximity bias reduction

For each chromosome arm, we first calculate the feature-wise mean across all unexpressed genes on that arm and then subtract those means from the features for each gene located on that arm. The gene locations were identified by National Center for Biotechnology Information RefSeq transcript locations against the hg38 reference assembly. The unexpressed genes were defined as those with $z$FPKM (fragments per kilobase of transcript per million mapped reads) $< -3.0$ in normalized bulk RNA sequencing of the given cell type before any CRISPR–Cas9 treatment.

## Reporting summary

Further information on research design is available in the Nature Portfolio Reporting Summary linked to this article.

## Data availability

Raw images, metadata and deep-learning-derived embeddings for rxrx3 are available at https://rxrx.ai; however, the majority of gene identities are currently masked due to commercial considerations. Due to contractual obligations with partners, Recursion is unable to share additional data underlying the rxrx3 analyses or the bulk RNA sequencing in Fig. 2f. cpg0016 is available as part of the JUMP Cell Painting datasets available from the Cell Painting Gallery on the Registry of Open Data on Amazon Web Services at https://registry.opendata.aws/cellpainting-gallery/. The scRNA-seq datasets are available through scPerturb (https://scperturb.org/). The DepMap data are available at https://depmap.org/portal/download/all/. The JUMP CP data were downloaded from S3 using python (1.22.3) and pandas (1.4.2) at https://registry.opendata.aws/cellpainting-gallery/. The hg38 gene locations and annotations were downloaded from the University of California, Santa Cruz Genome Browser at https://genome.ucsc.edu/cgi-bin/hgTables. The shinyDepMap data were downloaded and processed using R (v4.1) at https://depmap.org/portal/download/all/. The files containing scRNA-seq AnnData objects for two CRISPR–Cas9 and three CRISPRi screens were downloaded from Zenodo at https://zenodo.org/record/7416068 (ref. 74). Source data are provided with this paper.

## Code availability

The Python-based data analysis source code to reproduce plots from public datasets is available at https://github.com/recursionpharma/proxbias and Zenodo at https://doi.org/10.5281/zenodo.10795539 (ref. 75). Due to contractual obligations with partners, Recursion is unable to share code to reproduce plots from the rxrx3 data or Fig. 2f.

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

## Acknowledgements

We thank G. H. L. Roberts, H. Donnella, S. Guler, T. Ahfeldt, Y. Chong and K. Thomas for their help and discussions in generating this manuscript and the incredible Recursion lab and engineering teams for design and execution of experiments and storage and processing of data.

## Author contributions

I.S.H. contributed to the writing—initial draft. All authors contributed to the writing, review and editing. N.H.L., C.A.T., I.S.H., S.C., J.C.I., L.C., J.J., J.U., M.M.F. and W.P.B. contributed to the formal analysis, visualization, validation and methodology. C.C.G. and I.S.H. contributed to the supervision.

## Competing interests

All authors are current or former employees of Recursion Pharmaceuticals, Inc., and have received real or optional ownership interest in the company.

## Additional information

**Extended data** is available for this paper at https://doi.org/10.1038/s41588-024-01758-y.

**Correspondence and requests for materials** should be addressed to Imran S. Haque.

**A**

| vATPase | Autophagy | EGFR | Dynein | TGFB/Activin | Proteasome | Exosome | Replication | RNA |
|---|---|---|---|---|---|---|---|---|
| ATP6V1A | ATG12 | SHC1 | DYNC1I2 | ACVR1B | PSMA7 | DIS3 | RFC3 | POLA2 |
| ATP6V1B2 | ATG5 | HRAS | DYNC1H1 | TGFBR2 | PSMA5 | EXOSC4 | RFC2 | POLA1 |
| ATP6V1D | | PRKCE | DYNC1LI1 | TGFBR1 | PSMA3 | EXOSC8 | RFC4 | POLR2L |
| ATP6V1F | | BRAF | DYNC1LI2 | | PSMB3 | EXOSC9 | RFC5 | POLR2B |
| ATP6V1H | | EGFR | | | PSMB4 | EXOSC7 | | POLR2I |
| ATP6V1E1 | | RAF1 | | | PSMA6 | EXOSC5 | | POLR2G |
| | | MAPK1 | | | PSMA4 | | | POLR2C |
| | | | | | PSMB6 | | | |
| | | | | | PSMB1 | | | |
| | | | | | PSMA1 | | | |
| | | | | | PSMA2 | | | |
| | | | | | PSMB7 | | | |

**B**

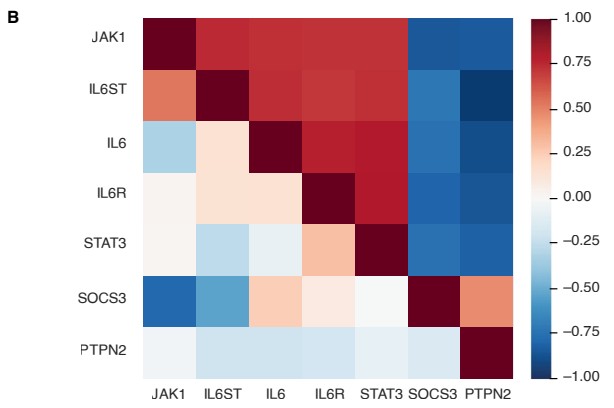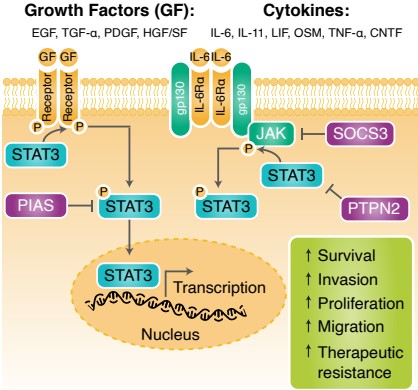

**C**

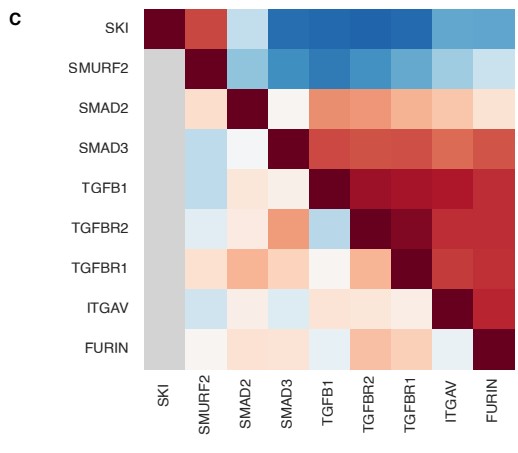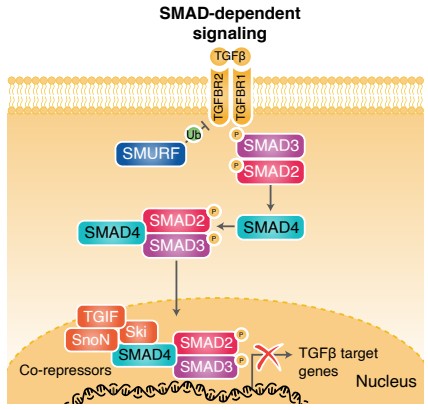

**D**

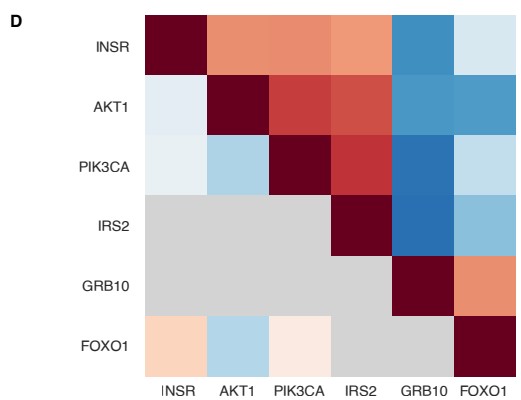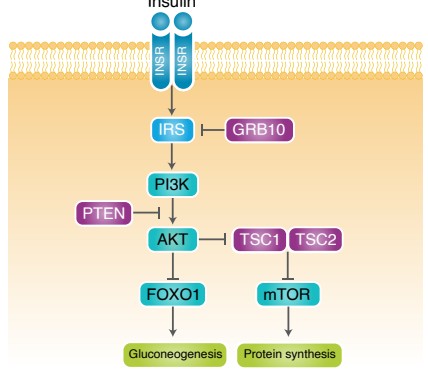

**Extended Data Fig. 1 | See next page for caption.**

**Extended Data Fig. 1 | Rxrx3 and cpg0016 example pathways. a**, Table of genes shown in Fig. 1b. **b, c, d**, Heatmaps of rxrx3 (above diagonal) and cpg0016 (below diagonal) data for selected biological pathways with corresponding pathway diagrams for JAK/STAT, TGF-beta, and insulin biology. Data not present in cpg0016 shown in gray. **b**, Example interleukin (IL) 6 pathway: *IL6*, *IL6R*, *IL6ST*, *JAK1*, and *STAT3* activate the IL-6 signaling pathway[76]. CRISPR-Cas9 targeting of these genes leads to similar cellular phenotypes and produces positive cosine similarities (red squares between *IL6*, *IL6R*, *IL6ST*, *JAK1*, and *STAT3* in the heatmap). As inhibitors of the IL-6 pathway, *SOCS3* and *PTPN2* demonstrate a negative cosine similarity to the pathway components *IL6/IL6R/IL6ST/JAK1/STAT3*, especially in the rxrx3 HUVEC data (blue squares)[76,77]. **c**, Example TGF-beta pathway: *FURIN*, *TGFB1*, *TGFBR1*, *TGFBR2*, *SMAD2*, and *SMAD3* activate the

TGF-beta pathway. CRISPR-Cas9 targeting of these genes gives a similar cellular phenotype and high cosine similarity (red squares in the heatmap)[78]. *SMURF2* and *SKI* inhibit the TGF-beta pathway, and CRISPR-Cas9 targeting of *SMURF2* and *SKI*, show high cosine similarity to each other but negative cosine values to *FURIN*, *TGFB1*, *TGFBR1*, *TGFBR2*, *SMAD2*, and *SMAD3* (blue squares)[78,79]. Grey squares indicate genes not present in the cpg0016 data. **d**, Example insulin pathway: *INSR*, *IRS2*, *AKT1*, *PIK3CA* transmit insulin signaling[80]. CRISPR-Cas9 targeting these factors gives similar phenotypes and therefore they are highly cosine-similar (red squares between *INSR/IRS2/AKT1/PIK3CA* in the heatmap). *GRB10* and *FOXO1* inhibit insulin signaling, reflected in negative cosine similarities between CRISPR-Cas9 targeting of *GRB10* and *INSR*, *IRS2*, *AKT1* and *PIK3CA* (blue squares)[80]. Grey squares indicate genes not present in the cpg0016 data.

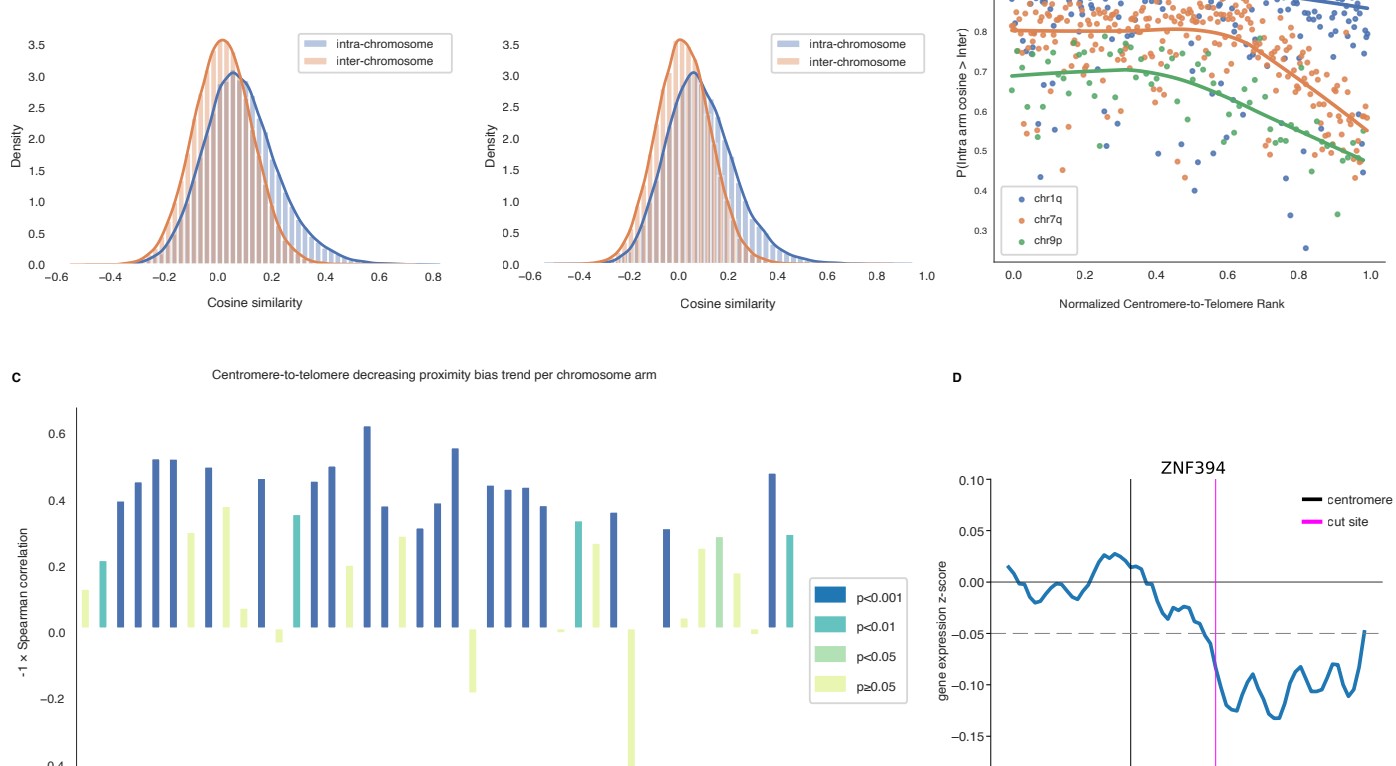

**Extended Data Fig. 2 | Additional figures showing proximity bias effects in rxrx3 and cpg0016. a**, Distribution plots of within-chromosome and between-chromosome cosine similarity for rxrx3 (left) and cpg0016 data (right). The within-chromosome distribution is shifted toward the positive, which was the initial indication that some bias was present. **b**, Scatterplot of gene-level one-sided Brunner-Munzel probabilities versus relative chromosome-arm position for three chromosome arms in the cpg0016 dataset. The value on the y-axis estimates the probability of an intra-chromosome-arm relationship involving

a given gene having a higher cosine similarity than an inter-chromosome-arm relationship involving the same gene. **c**, Spearman correlations in plots similar to b across all chromosome arms for the cpg0016 data. The height of the bar for each arm agrees well with the degree of fading in diagonal blocks in Fig. 1c below the diagonal. Colors show Bonferroni-corrected *p*-values. **d**, Bulk RNA sequencing gene count depletion for cells treated with a ZNF394-targeting guide relative to untreated cells in 10-gene blocks across chromosome 7. Decreased expression is evident on the telomeric side of the cut site.

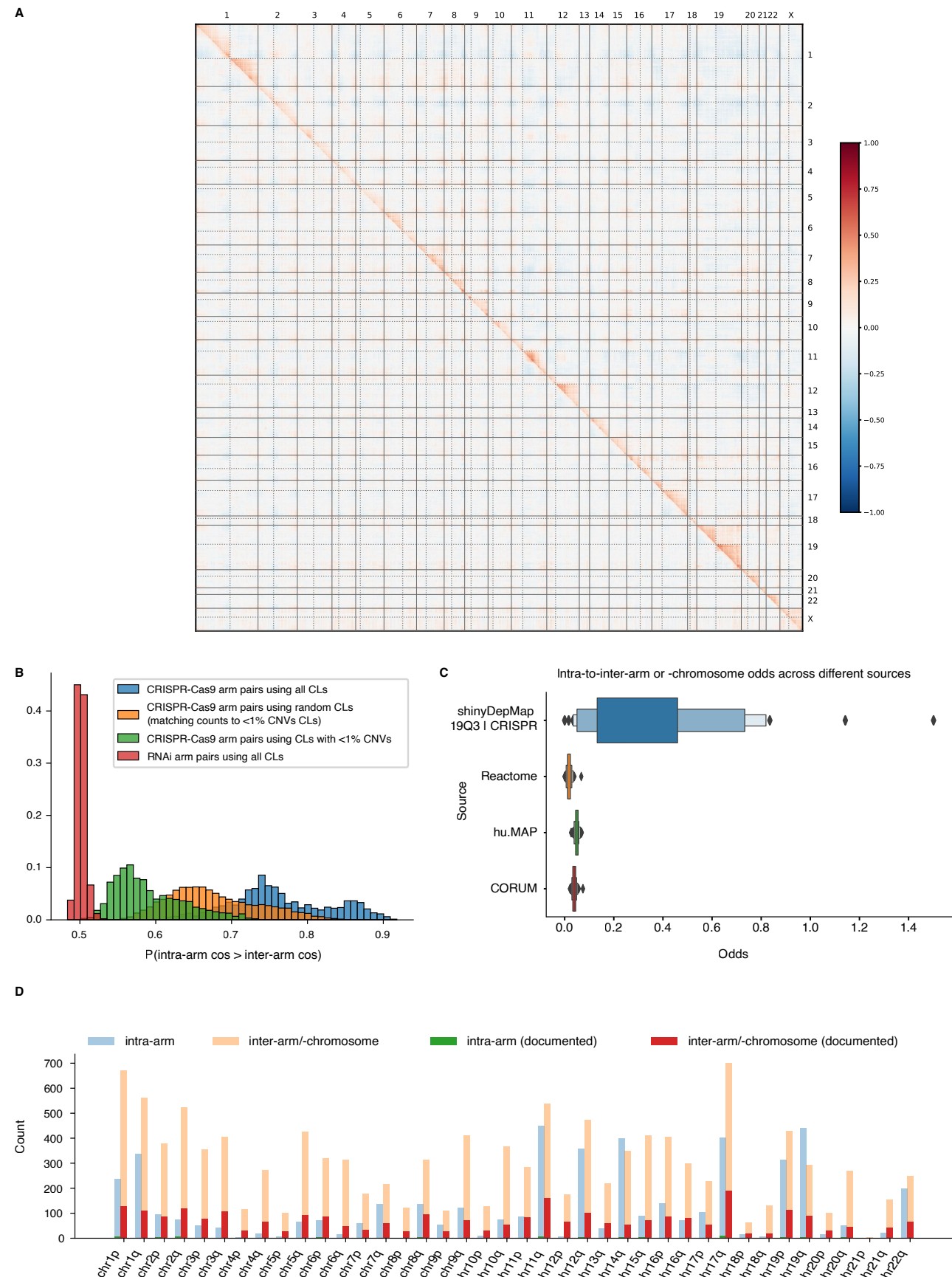

**Extended Data Fig. 3 | See next page for caption.**

**Extended Data Fig. 3 | Proximity bias quantification in DepMap data. a**, Split genome-wide heatmap of the DepMap 22Q4 (above diagonal) and 23Q2 (below diagonal) CRISPR data. Both are processed with the Chronos pipeline[17] but 23Q2 has an additional correction applied to reduce proximity bias. 22Q4 has 1,078 cell lines, 23Q2 has 1,095 cell lines. **b**, Distributions of arm-level Brunner-Munzel probabilities for maps built using pairs of autosomal chromosome arms (741 pairs represented twice in blue, green and red distributions). Blue distribution is built using all DepMap 22Q4 CRISPR-Cas9 cell lines, orange samples random cell lines matching the numbers from the green distribution (10 random sampling runs), green uses only cell lines with less than 1% of genes having copy number calls outside of [1.75, 2.25] (counts in Supplementary Table 4), and red uses all cell lines in the DepMap shRNA data. Two-sided Mann-Whitney U tests between all distributions are highly significant (p-value < 1E-10) for all pairwise comparisons. **c**, Boxen plots showing distributions of the ratio of within-chromosome-arm relationships to between-arm relationships for each chromosome arm across different gene annotation sets (n = 39 chromosome arms for all sources. Boxes are drawn at each octile with outliers outside of those boxes. The *19Q3* DepMap data show a much higher ratio of within-arm to between-arm annotations, suggesting a systematic bias to the predicted associations. **d**, Counts of gene-gene relationships within and between chromosome arms for shinyDepMap 19Q3 data (blue and tan, n = 4747 and 9271 respectively) and public annotation sets (Reactome, HuMap, and CORUM) (green and red, n = 98 and 2825 respectively)[25-27]. DepMap predicts a much higher proportion of within-chromosome-arm relationships than are found in public annotation sets (odds ratio 0.068, Fisher exact p-value < 1e-10).

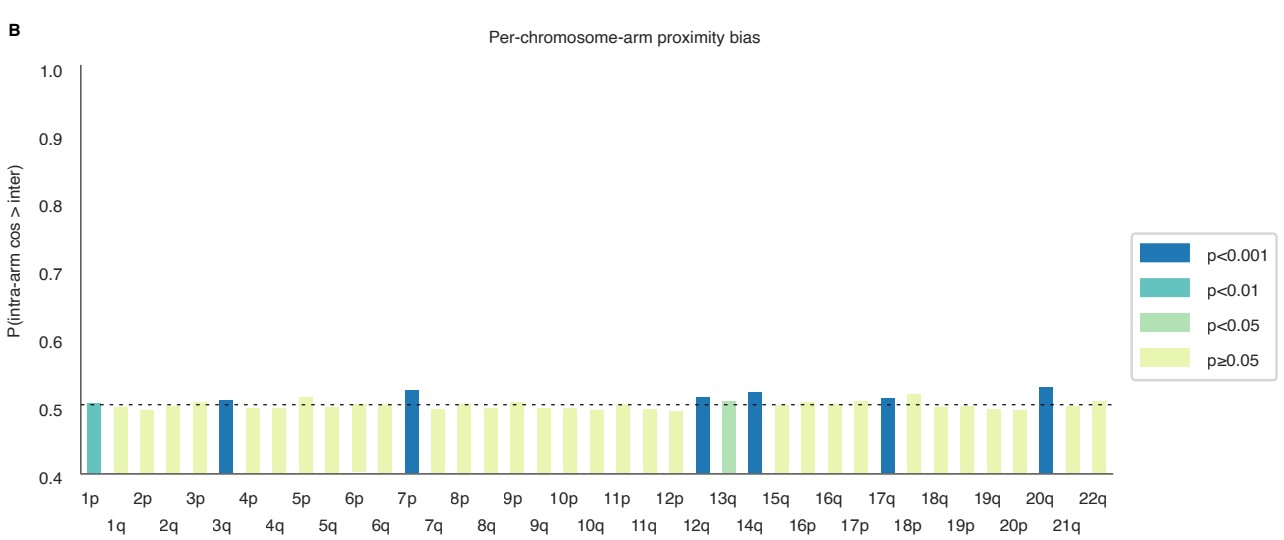

**Extended Data Fig. 4 | See next page for caption.**

**Extended Data Fig. 4 | Proximity bias correction in DepMap Data. a**, Split genome-wide heatmap built from 625 CRISPR cell lines, 190 shRNA cell lines, and 11,169 genes shared between CRISPR and shRNA datasets in the DepMap 19Q3 and DEMETER2 v6 data. CRISPR-Cas9 data are shown above the diagonal and shRNA data below. No proximity bias signal is visible in the shRNA data.

b, Quantification of proximity bias in the DepMap shRNA dataset with colors showing Bonferroni-corrected p-values from the one-sided arm-level Brunner-Munzel test. Only a few chromosome arms display significant deviation of intra-versus inter-chromosome-arm similarities contrasting with the CRISPR data shown in Fig. 3b.

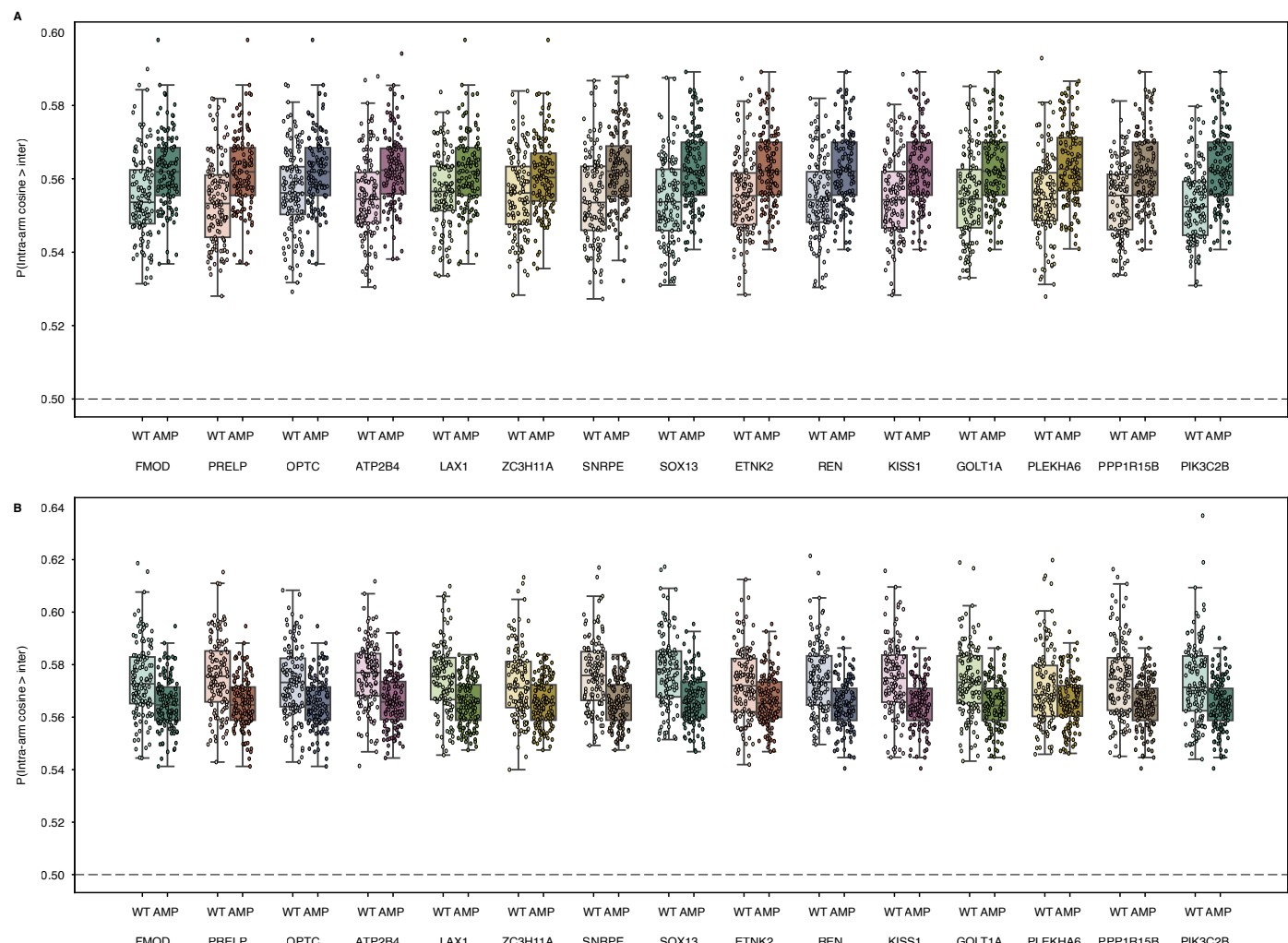

**Extended Data Fig. 5 | Proximity bias quantification for additional genes between *BTG2* and *MDM4*.** *Box and scatter plots of whole-genome level proximity bias quantification by Brunner-Munzel intra-arm vs inter-arm probability from DepMap 22Q4 data with cell lines stratified by gene status. Each point represents a bootstrap sample of cell lines, 128 bootstraps were run for each condition. Box plots show the median, lower and upper quartile with whiskers extending to the furthest points within 1.5 times the inner quartile range. Detailed test statistics in Supplementary Table 3.* **a**, All genes with sufficient data in order of chromosome position between *BTG2* and *MDM4* on chromosome 1q; wild-type (WT) vs amplification (AMP) in *TP53* WT background (*FMOD*: WT n = 175, AMP n = 87, *PRELP*: WT n = 174, AMP n = 87, *OPTC*: WT n = 172, AMP n = 87, *ATP2B4*: WT n = 167, AMP n = 87, *LAX1*: WT n = 171, AMP n = 87, *ZC3H11A*: WT n = 172, AMP n = 87, *SNRPE*: WT n = 177, AMP n = 86, *SOX13*: WT n = 169, AMP n = 87, *ETNK2*: WT n = 170, AMP n = 87, *REN*: WT n = 173, AMP n = 87, *KISS1*: WT n = 176, AMP n = 87, *GOLT1A*: WT n = 173, AMP n = 87, *PLEKHA6*: WT n = 158, AMP n = 86, *PPP1R15B*: WT n = 171, AMP n = 87, *PIK3C2B*: WT n = 158, AMP n = 87). **b**, Same as a, in *TP53* LOF background (*FMOD*: WT n = 182, AMP n = 81, *PRELP*: WT n = 183, AMP n = 81, *OPTC*: WT n = 182, AMP n = 81, *ATP2B4*: WT n = 180, AMP n = 80, *LAX1*: WT n = 184, AMP n = 80, *ZC3H11A*: WT n = 182, AMP n = 80, *SNRPE*: WT n = 189, AMP n = 80, *SOX13*: WT n = 180, AMP n = 80, *ETNK2*: WT n = 186, AMP n = 80, *REN*: WT n = 186, AMP n = 79, *KISS1*: WT n = 189, AMP n = 79, *GOLT1A*: WT n = 185, AMP n = 79, *PLEKHA6*: WT n = 174, AMP n = 78, *PPP1R15B*: WT n = 178, AMP n = 79, *PIK3C2B*: WT n = 172, AMP n = 79). Genes with less than 25 cell lines in a given condition are not shown.

# Reporting Summary

## Statistics

For all statistical analyses, confirm that the following items are present in the figure legend, table legend, main text, or Methods section.

| n/a | Confirmed | |
|---|---|---|
| ☐ | ☒ | The exact sample size (*n*) for each experimental group/condition, given as a discrete number and unit of measurement |
| ☐ | ☒ | A statement on whether measurements were taken from distinct samples or whether the same sample was measured repeatedly |
| ☐ | ☒ | The statistical test(s) used AND whether they are one- or two-sided<br>*Only common tests should be described solely by name; describe more complex techniques in the Methods section.* |
| ☒ | ☐ | A description of all covariates tested |
| ☐ | ☒ | A description of any assumptions or corrections, such as tests of normality and adjustment for multiple comparisons |
| ☐ | ☒ | A full description of the statistical parameters including central tendency (e.g. means) or other basic estimates (e.g. regression coefficient) AND variation (e.g. standard deviation) or associated estimates of uncertainty (e.g. confidence intervals) |
| ☐ | ☒ | For null hypothesis testing, the test statistic (e.g. *F*, *t*, *r*) with confidence intervals, effect sizes, degrees of freedom and *P* value noted<br>*Give P values as exact values whenever suitable.* |
| ☒ | ☐ | For Bayesian analysis, information on the choice of priors and Markov chain Monte Carlo settings |
| ☒ | ☐ | For hierarchical and complex designs, identification of the appropriate level for tests and full reporting of outcomes |
| ☐ | ☒ | Estimates of effect sizes (e.g. Cohen's *d*, Pearson's *r*), indicating how they were calculated |

*Our web collection on statistics for biologists contains articles on many of the points above.*

## Software and code

Policy information about availability of computer code

| | |
|---|---|
| Data collection | Data were downloaded and processed with python (v3.9), numpy (v1.22.3) and pandas (1.4.2) and R (v4.1) |
| Data analysis | Analysis was performed using Python (v3.9) and open-source packages numpy(v1.22.3), pandas(v1.4.2), scipy (v1.10.1) and statsmodels (v0.13.2). Visualizations generated with matplotlib (v3.5.2), scikit-image (v0.18.3), seaborn (v0.12.3). single-cell RNA sequencing data was processed with STAR (v2.7.7a) and scanpy (v1.9.3) and determined the CNV events using infercnvpy (v0.4.1). The list of cancer genes was downloaded from OncoKB (v4.4).<br>Custom code available at https://github.com/recursionpharma/proxbias/ |

For manuscripts utilizing custom algorithms or software that are central to the research but not yet described in published literature, software must be made available to editors and reviewers. We strongly encourage code deposition in a community repository (e.g. GitHub). See the Nature Portfolio guidelines for submitting code & software for further information.

## Data

Policy information about availability of data

All manuscripts must include a data availability statement. This statement should provide the following information, where applicable:

- Accession codes, unique identifiers, or web links for publicly available datasets
- A description of any restrictions on data availability
- For clinical datasets or third party data, please ensure that the statement adheres to our policy

Raw images, metadata, and deep-learning-derived embeddings for rxrx3 are available at https://rxrx.ai.  cpg0016 is available as part of the JUMP Cell Painting datasets available from the Cell Painting Gallery at https://registry.opendata.aws/cellpainting-gallery/.  DepMap data are available at https://depmap.org/portal/download/all/ JUMP CP data was downloaded from their S3 bucket using python (1.22.3) and pandas (1.4.2): https://registry.opendata.aws/cellpainting-gallery/. hg38 gene locations and annotations were downloaded from UCSC: https://genome.ucsc.edu/cgi-bin/hgTables shinyDepMap data was downloaded and processed using R (4.1): https://depmap.org/portal/download/all/ Files containing scRNAseq AnnData objects for two CRISPR-Cas9 and three CRISPRi screens were downloaded from zenodo.org/record/7416068. The list of cancer genes was downloaded from OncoKB (v4.4), https://www.oncokb.org/cancer-genes.

## Research involving human participants, their data, or biological material

Policy information about studies with human participants or human data. See also policy information about sex, gender (identity/presentation), and sexual orientation and race, ethnicity and racism.

| | |
|---|---|
| Reporting on sex and gender | The HUVEC source cells used in the rxrx3 dataset were sourced from two donor pools.<br>1) Lonza Catalog # C2519A Lot 0000662339 Manufacture Date 26-Sept-2017 (3 pooled umbilical cord donors) Age = Newborn, Sex = Male, Male, Male, Race = Caucasian, Caucasian, Caucasian<br>2) Lonza Catalog # C2519A Lot 0000661173 Manufacture Date 22-Sept-2017 (6 pooled umbilical cord donors)  Age = Newborn, Sex = Male, Female Mixed, Race = Black, Other, Caucasian, Caucasian, Black, Caucasian The cells were isolated from donated human tissue after obtaining permission for use in research applications by informed consent or legal authorization. |
| Reporting on race, ethnicity, or other socially relevant groupings | No reference to race, ethnicity or other social groups is made in the manuscript. |
| Population characteristics | See above |
| Recruitment | No recruitment was performed. |
| Ethics oversight | No ethics oversight body was consulted. |

Note that full information on the approval of the study protocol must also be provided in the manuscript.

# Field-specific reporting

Please select the one below that is the best fit for your research. If you are not sure, read the appropriate sections before making your selection.

☒ Life sciences  ☐ Behavioural & social sciences  ☐ Ecological, evolutionary & environmental sciences

For a reference copy of the document with all sections, see [nature.com/documents/nr-reporting-summary-flat.pdf](http://nature.com/documents/nr-reporting-summary-flat.pdf)

# Life sciences study design

All studies must disclose on these points even when the disclosure is negative.

| | |
|---|---|
| Sample size | No statistical method was used to predetermine sample size. |
| Data exclusions | No data were excluded. |
| Replication | Major findings were replicated across the rxrx3 and cpg0016 datasets as well as within three data sources from DepMap (19Q3, 22Q4, and 23Q2). |
| Randomization | No interventional analyses were performed as a part of this study, so no randomization of samples into experimental groups was used. |
| Blinding | Blinding is not relevant to this study because there were no interventions applied that could be influenced by experimenters. |

# Reporting for specific materials, systems and methods

We require information from authors about some types of materials, experimental systems and methods used in many studies. Here, indicate whether each material, system or method listed is relevant to your study. If you are not sure if a list item applies to your research, read the appropriate section before selecting a response.

## Materials & experimental systems

| n/a | Involved in the study |
|-----|------------------------|
| ☒ ☐ | Antibodies |
| ☐ ☒ | Eukaryotic cell lines |
| ☒ ☐ | Palaeontology and archaeology |
| ☒ ☐ | Animals and other organisms |
| ☒ ☐ | Clinical data |
| ☒ ☐ | Dual use research of concern |
| ☒ ☐ | Plants |

## Methods

| n/a | Involved in the study |
|-----|------------------------|
| ☒ ☐ | ChIP-seq |
| ☒ ☐ | Flow cytometry |
| ☒ ☐ | MRI-based neuroimaging |

## Eukaryotic cell lines

Policy information about cell lines and Sex and Gender in Research

| Cell line source(s) | The HUVEC source cells used in the rxrx3 dataset were sourced from two donor pools.<br>1) Lonza Catalog # C2519A Lot 0000662339 Manufacture Date 26-Sept-2017 (3 pooled umbilical cord donors)<br>Age = Newborn, Sex = Male, Male, Male, Race = Caucasian, Caucasian, Caucasian<br>2) Lonza Catalog # C2519A Lot 0000661173 Manufacture Date 22-Sept-2017 (6 pooled umbilical cord donors)<br>Age = Newborn, Sex = Male, Female Mixed, Race = Black, Other, Caucasian, Caucasian, Black, Caucasian<br>The cells were isolated from donated human tissue after obtaining permission for use in research applications by informed consent or legal authorization. |
|---|---|
| Authentication | Cell lines were only utilized from public data sets (cpg0016 and DepMap). |
| Mycoplasma contamination | Cell lines were only utilized from public data sets (cpg0016 and DepMap). |
| Commonly misidentified lines<br>(See ICLAC register) | Cell lines were only utilized from public data sets (cpg0016 and DepMap). |

