## [Peer Review File · Nature Genetics]

Peer Review Information

Manuscript Title: High-resolution genome-wide mapping of chromosome-arm-scale truncations induced by CRISPR-Cas9 editing

Corresponding author name(s): Dr Imran Haque

Reviewer Comments & Decisions:

Decision Letter, initial version:
--

21st Aug 2023

Dear Dr Haque,

Your Article, "High-resolution genome-wide mapping of chromosome-arm-scale truncations induced by CRISPR-Cas9 editing" has now been seen by 3 referees. You will see from their comments copied below that while they find your work of considerable potential interest, they have raised quite substantial concerns that must be addressed. In light of these comments, we cannot accept the manuscript for publication, but would be very interested in considering a revised version that addresses these serious concerns.

We hope you will find the referees' comments useful as you decide how to proceed. If you wish to submit a substantially revised manuscript, please bear in mind that we will be reluctant to approach the referees again in the absence of major revisions.

To guide the scope of the revisions, the editors discuss the referee reports in detail within the team, including with the chief editor, with a view to identifying key priorities that should be addressed in revision and sometimes overruling referee requests that are deemed beyond the scope of the current study. In this case we would like you to address Reviewers' comments in full. Particularly, we think it would be important to assess the prevalence and functional impact of the proximity bias. Please do not hesitate to get in touch if you would like to discuss these issues further.

If you choose to revise your manuscript taking into account all reviewer and editor comments, please highlight all changes in the manuscript text file. At this stage we will need you to upload a copy of the manuscript in MS Word .docx or similar editable format.

*2) If you have not done so already please begin to revise your manuscript so that it conforms to our Article format instructions, available here. Refer also to any guidelines provided in this letter.

Please be aware of our guidelines on digital image standards.

[redacted]

If you wish to submit a suitably revised manuscript we would hope to receive it within 6 months. If you cannot send it within this time, please let us know. We will be happy to consider your revision so long as nothing similar has been accepted for publication at Nature Genetics or published elsewhere. Should your manuscript be substantially delayed without notifying us in advance and your article is eventually published, the received date would be that of the revised, not the original, version.

Thank you for the opportunity to review your work.

Sincerely,
Chiara

Chiara Anania, PhD
Associate Editor
Nature Genetics
<https://orcid.org/0000-0003-1549-4157>

Referee expertise:

Referee #1: cell division/genome instability

Referee #2: genetics/genomic instability/cancer

Referee #3: computational Biology/bioinformatics/cancer genomics

Reviewers' Comments:

Reviewer #1:

Remarks to the Author:

Lazar et al. describes and offers a computational correction to a previously described unintended consequence of CRISPR-Cas9 treatment—chromosome arm loss from the target site to the telomere—in the unique context of CRISPR screens. They also assess the pathways that contribute to arm loss in the context of CRISPR cuts. They find this result within a phenomics imaging screen of primary HUVECs, and back up their results with previous screens performed by others under different conditions. They also perform an analysis of DepMap data to find that cell cycle-related genes contribute to this artifact.

Although it has been well documented that CRISPR/Cas9 can lead to chromosome truncations (Cullot et al., 2018; Zuccaro et al., 2020; Alanis-Lobato et al., 2021; Leibowitz et al., 2021; Sheltzer et al., 2021, and others), no group has published this leading to artifacts in CRISPR screens. Some groups have found fitness similarities amongst genes within the same chromosome in CRISPR screens that are likely due to the artifact described by Lazar et al., (Amici et al., 2020), but did not identify chromosome truncation after cutting as the cause of the artifact.

Despite previous knowledge that chromosome missegregation can lead to arm-level truncations, the unique context of this result and the authors' generation of a correction for it within screening data render the manuscript interesting to those performing CRISPR screening of any sort. The work also appears robust. The interest, however, should be caveated with the fact that these artifacts are relatively rare, as illustrated by this paper and other previous papers. There remain several points that should be addressed:

(1) MDM4 amplification should phenocopy TP53 loss, leading to increased cycling, and one would expect high rates of arm loss/chromosome missegregation. However, this is the opposite of the result observed. One explanation for this could be faulty annotation of P53 LOF (either by the authors or due to faulty DepMap data) wherein the true comparison being shown here is P53 loss (as MDM4 WT) versus P53 WT (as MDM4 GOF). Is there a way to test for this or control this? One way could be to show the effect of MDM4 amplification in putative P53 WT cells where artifacts in P53 loss are less

likely. If MDM4 amplification in this case gives the opposite result then this result requires further refinement. If it gives the same result, then discussion is warranted for the molecular causes of this result.

(2) It is apparent in Fig 1D (e.g. chr1), Fig 2a, S2b, that some chromosome arms have inflection points where the likelihood of having a truncation event is dramatically changed. Discussion should be added as to possible causes of this. Data determining whether these sites are at loci containing either haploinsufficient essential genes or other genes that cause gross changes in phenotype would also be useful towards interpretation of these inflection points.

(3) In Table 1 and S2, there seems to be a high rate of arm loss centromeric to the cut site as well as telomeric. First, please include whether there is there a statistically significant enrichment of telomeric loss (by Fisher's Exact test or similar). Next, in discussion or elsewhere, please include what is the explanation for this. In the discussion you seem to invoke acrocentric chromosomes, however many chromosomes with centromeric loss are not acrocentric. Are genes with loss towards the centromere nearer the centromere than telomere? Relatedly, a discussion of this area would be useful. Are these due to segmental losses as seen in Kosicki et al., 2018? Chromosome bridges as in Leibowitz et al.?

(4) The authors of the manuscript should be more careful with language around significance so as to not confuse it with statistical significance. In line 217 there is a suggestion of a significant change of proximity bias depending upon P53 status. While an effect is apparent in the associated figures, I do not see a test for the statistical significance of the comparison of P53 WT vs LOF. Similar analyses or softening of language is appropriate for other genes as well.

(5) More care should be taken to cite papers appropriately. There is no citation to previous work looking at coessentiality on genes (namely Amici et al., 2020), or to work that suggests cell cycle/p53 as a cause for truncation (Cullot et al., 2023). Moreover, although citations are present for the appropriate literature in general, the authors should cite them for their main contributes more clearly. For example, Kosicki et al found kilobase scale deletions, but is not cited for this in the introduction (line 43), Zuccaro et al. (2020) and Leibowitz et al., (2021) both demonstrate chromosome missegregation as the cause for truncation events consistent with your paragraph beginning on line 291.

(6) It would be prudent to soften some conclusion language. For example, Add the word "potentially" to line 252 (Potentially outweighed).

(7) It is worth finishing the paragraph at line 290 with a statement that no negative consequences due to unintended effects of CRISPR cutting have been seen in patients, potentially mitigating the risk of unintended loss.

Reviewer #2:

Remarks to the Author:

In this manuscript, the authors describe an imaging-based phenotypic profiling approach to characterize the consequences of CRISPR-Cas9 editing. They performed a genome-wide CRISPR-Cas9 screen in a human cancer cell line, using cell imaging as their readout (which they call "phenomics"), and calculated the pairwise similarity of gene KO. They found that genes on the same chromosome-

arm exhibit more similar KO profiles (in comparison to random pairs). Transcriptomic data analysis showed loss of chromosome regions telomeric to the cut sites of target genes, suggesting that chromosome truncation may underlie this "proximity bias". The chromosome-arm proximity bias affected genome-wide CRISPR screening, as gene-gene essentiality relationship was more likely to be detected for genes that reside on the same chromosome-arm (relative to that expected based on databases of known biological relationships). The proximity effect was specific to CRISPR-Cas9 genome editing, and not observed in CRISPRi or shRNA screens. A simple chromosome-arm normalization significantly reduced this effect.

The observation that gene essentiality in CRISPR-Cas9 screens is affected by gene proximity is interesting and of value for the field. However, it's not quite as novel as presented in the manuscript. The suggestion that this "proximity bias" is driven by chromosome-arm truncations is also intriguing, but the truncations themselves (and their relationship with p53) have been established before, and the specific mechanistic link to the observed "proximity bias" is not sufficiently proven in this manuscript. Overall, while the phenomenon described in the manuscript is certainly interesting, some aspects of the manuscript are not sufficiently novel, and others are novel but not fully proven by the presented analyses.

Major comments

(1) The manuscript fails to cite/discuss several important and relevant papers, which somewhat diminish the novelty of several aspects of the study. These include:

(a) Papers discussing the non-specific DNA damage and p53 activation following CRISPR-Cas9 genome editing (Haapaniemi et al. Nat Med 2018; Ihry et al. Nat Med 2018; Enache et al. Nat Genet 2020), and its effect on genome-wide screens (Enache et al. Nat Genet 2020; Bowden et al. eLife 2020, Sinha et al. Nat Commun 2021)

(b) Papers discussing the effect of DNA copy-number changes on CRISPR-Cas9 editing toxicity and genome-wide screens and its computational correction (Aguirre et al. Cancer Discov 2016; Meyers et al. Nat Genet 2017)

(c) A recent paper that showed chromosome-arm copy number losses, as well as deletions between the target cut site and the telomere, following CRISPR-Cas9 editing in human T cells (Nahmad et al. Nat Biotechnol 2022). As reported in the current manuscript, that paper used scRNAseq to identify CRISPR-Cas9 chromosome truncations telomeric to the targeted genes.

These papers should be cited and discussed properly.

(2) To what extent does the "proximity bias" reflect the copy-number effect (described in Aguirre et al. and Meyers et al.)? This may be a major contributor both to the phenotypic similarity that is reported, and (especially) to the observed effect on CRISPR-Cas9 screens, but this is not analyzed and not discussed in this manuscript. While the observation of the "proximity blocks" in karyotypically-normal human primary cells indicates that the effect could not be explained in its entirety by CNAs, CNAs may still be a major contributor to the effect of this "proximity bias" in cancer cell line screens. A thorough analysis is required to separate between these two closely related effects.

(3) In line with the previous point, the authors report that the chromosome-arm proximity bias was partially corrected in the 23Q2 version of DepMap (Supplementary Fig. 3), which introduced Chronos 2.0 and changed a few technical details in the analysis of data from CRISPR screens. From a practical standpoint, this suggests that the problem has been solved in the DepMap dataset, attenuating the need for the correction proposed at the end of the manuscript. More importantly, can the authors

comment on why the 23Q2 changes attenuated the effect so well, and in what ways the correction that they propose is different/better?

(4) TP53 status has been shown to affect CRISPR-Cas9 screens (Enache et al. Nat Genet 2020; Bowden et al. eLife 2020, Sinha et al. Nat Commun 2021). The authors report a p53-dependent effect on the "proximity bias". This again might be related to the copy number effect, as TP53-deficient cell lines would carry many more CNAs. And, again, it raises the question of the extent to which the "proximity bias" effect overlaps the previously reported effect of p53 on CRISPR screens. This should be evaluated by a direct comparison.

(5) The "mechanistic model" for the generation of chromosomal truncations by CRISPR-Cas9 (Fig. 4f) is particularly weak, as it is not supported by substantial data. While the model itself makes sense and fits previous reports from the literature (Nahmad et al. Nat Biotechnol 2022; Tsuchida et al. bioRxiv 2023), the current manuscript provides very little additional support for the model. This part of the Results would fit the Discussion section better than the Results section.

(6) Can the authors compare the imaging-based phenotypes that they identified to those from previous CRISPR screens that used high-content imaging as a readout (e.g., Yan et al. JCB 2021)? If so, does the "proximity bias" exist in those additional datasets as well?

Reviewer #3:

Remarks to the Author:

The manuscript entitled "High-resolution genome-wide mapping of chromosome-arm-scale truncations induced by CRISPR-Cas9 editing" by Lazar et al reports that unrelated genomically proximal genes located on the same chromosome arm are systematically more similar than expected in CRISPR-Cas9 screening. This phenomena, termed "proximity bias", was initially discovered using data generated from cellular morphological profiling of two cell lines subjected to CRISPR-Cas9 screening: human umbilical vein endothelial cells by the authors and the U2OS cells by the Joint Undertaking in Morphological Profiling-Cell Painting consortium. Proximity bias was also found in DepMap (Dependency Map) data. Prior publication and CNV analysis of the scRNA-seq generated from CRISPR-CAS9 screening indicate that this is likely caused by loss of chromosome arm close to the double-stranded breaks introduced by CAS9 nuclease and had the strongest effect on genes closest to centromere. Proximity bias appears to be affected by cell cycle activity, in particular, TP53 LOF status, which is consistent with prior literature. The authors also showed that proximity bias could be mitigated by geometric correction by subtracting an estimated representation of the chromosome arm based on gene location.

The paper is of tremendous interest to the field as the effect of chromosomal truncation has not been well-appreciated in genetic screening. The study can have the potential influencing the future design of genetic screening and help improving analysis with data generated with suc bias. However, there are several major and minor issues related to data analysis and quality that need to be addressed.

Major issues:

1. Novelty on contribution of chromosome truncation/TP53 LOF to proximity bias. This has been reported by Tsuchida et al ("the Mitigation of chromosome loss in clinical CRISPR-Cas9-engineered T cells", ref 8) and the mechanism is very similar to what's described in Figure 4d. It is important to emphasize the aspect of replicating the finds made by Tsuchida et al.

2. A major concern of the proximity bias is the false positive rate on the dependency genes (target discovery) identified by CRISPR screen due to the proximity effect. Based on the data presented in this paper and the prior publication, chromosome loss only affects a small fraction of the cells (4.2%-15%). Can such a small fraction of cells lead to false discovery of dependency genes or just the general pattern of dependency scores? For example, Online Methods describes that the dependency scores for each gene were centered by subtraction the means of all cell lines to mitigate the effect of essential genes (lines 457-458). Additional analysis is needed to demonstrate how the proximity bias affects the dependency genes identified in each cell line, in particular those affected by TP53 LOF mutations. The general pattern presented in Figures 3 & 4 does not address this question. Illustration using false positive calls on a well-characterized dependency gene will be highly informative.
3. Dependency genes were identified by evaluating all cell lines and identify those that are specific to a subset of the cancer. Given proximity effect occurs universally in all cell types, will this (the analytical approaches for dependency genes) mitigate the effect of proximity on dependency gene discovery?
4. Geometric method for proximity bias reduction. The description in Online Methods was too general for evaluation and the resulting code needs to be made publicly available (The URL provided by the authors, <https://github.com/recursionpharma/proxbias>, is not accessible). In addition to improving the correlation, the authors need to show how the adjustment can reduce the potential false positives in the identification of essential genes in CRISPR screen.
5. RNA-seq analysis with the results presented in Table 1, Supplementary Table S2 and Figure 2 D-F.
 - a) The narrative on line 146-151 suggested 150 target genes flanking the target site were evaluated while the data presented in Table 1 is <150X2 genes. Please clarify this.
 - b) The proposed model indicating chromosome-arm level changes; however the results showed that only 8%-33% of targeted genes exhibited copy number loss, which is a surprise. Please clarify
 - c) Supplementary Table S2 also shows a high variability in % cells affect by copy number loss in the target genes. If the target genes are all within the chromosomal loss, we would expect to see consistent number of cells across. Please clarify the results. Bu contrast, CNV loss shown in Figure 2 looks uniform.
 - d) According to Online Methods "Analysis of public scRNAseq data", CNV analysis from scRNA-seq does not appear to be equivalent to target gene analysis, please clarify if that is the case. Was the data presented in Figure 2d and 2e based on inferCNV or it is based on the analysis described in Online Methods "An Analysis of public scRNAseq data"?
6. There are three different versions of DepMap data used, 19Q3, 22Q4 and 23Q2. Please describe the rational for using these different versions of data.

Minor issues:

1. Refs 1-3 do not appear to be relevant to chromosome truncations caused by CRISPR, which was the original intent. Instead, these appear to describe the clinical application of genetic engineering. Please make corrections by citing the appropriate references.
2. The content of Table 1 is duplicated in Supplementary Table 2.
3. The section of Online Methods "An Analysis of public scRNAseq data" describes both Table 1 and 2 in Line 433, however, Table 2 can not be found. Does this refer to Supplementary Table 2?
4. Figure 2 legend. The description of the CNV maps in panels d-f is mixed with that of panel C, a schematic drawing for the correlation model, which is quite confusing.
5. Line 843: 22Q2 should be relabeled as 23Q2
6. Line 179-182: Supplementary Figure S3 showed that DepMap 23Q2 has mitigated the effect of proximity bias by "by a correction similar to that described below". Is this a speculation? If not, please verify with the DepMap development team on how they reduced this effect in their new data release.

Author Rebuttal to Initial comments**Reviewer Responses**Reviewer 1:

Lazar et al. describes and offers a computational correction to a previously described unintended consequence of CRISPR-Cas9 treatment—chromosome arm loss from the target site to the telomere—in the unique context of CRISPR screens. They also assess the pathways that contribute to arm loss in the context of CRISPR cuts. They find this result within a phenomics imaging screen of primary HUVECs, and back up their results with previous screens performed by others under different conditions. They also perform an analysis of DepMap data to find that cell cycle-related genes contribute to this artifact.

Although it has been well documented that CRISPR/Cas9 can lead to chromosome truncations (Cullot et al., 2018; Zuccaro et al., 2020; Alanis-Lobato et al., 2021; Leibowitz et al., 2021; Sheltzer et al., 2021, and others), no group has published this leading to artifacts in CRISPR screens. Some groups have found fitness similarities amongst genes within the same chromosome in CRISPR screens that are likely due to the artifact described by Lazar et al., (Amici et al., 2020), but did not identify chromosome truncation after cutting as the cause of the artifact.

Despite previous knowledge that chromosome missegregation can lead to arm-level truncations, the unique context of this result and the authors' generation of a correction for it within screening data render the manuscript interesting to those performing CRISPR screening of any sort. The work also appears robust. The interest, however, should be caveated with the fact that these artifacts are relatively rare, as illustrated by this paper and other previous papers. There remain several points that should be addressed:

1. MDM4 amplification should phenocopy TP53 loss, leading to increased cycling, and one would expect high rates of arm loss/chromosome missegregation. However, this is the opposite of the result observed....Is there a way to test for this or control this?

We have revised this analysis.

We carefully re-analyzed our data and found that indeed amplification of *MDM4* in a *TP53* wild-type setting shows increased proximity bias. In our previous analysis the "GOF" condition included cell lines with nonsense, or frameshift indel mutations which are likely to create non-functional proteins. We have updated our process to exclude cell lines with deleterious

mutations and switched to using "AMP" to describe this condition (we still allow deleterious mutations in the LOF condition). Interestingly *MDM4* amplification shows differential behavior with respect to proximity bias in the *TP53* WT vs LOF background. In the presence of functional p53, *MDM4* amplification seems to exacerbate proximity bias effects, but with loss of p53, *MDM4* amplification reduces proximity bias. We have added discussion around this point for *MDM4* and several other genes.

Note that we require at least 25 cell lines in order to evaluate a gene in a given setting and we didn't meet that criteria for *MDM4* LOF in *TP53* WT or LOF backgrounds (3 and 10 cell lines respectively).

Additionally, the barplots in Figure 4 and Supplemental Figure 5 in the previous version did not control for the number of cell lines in each condition. We have updated those figures to show proximity bias metrics from a bootstrap sampling approach used to identify driver genes where the number of cell lines in each condition is held constant. In this process we randomly select 20 cell lines for each condition, create maps and calculate Brunner-Munzel probabilities in each bootstrap iteration. We then compare conditions using t-tests across 32 bootstraps for all cancer genes and confirm with 128 bootstrap samples for selected genes. The manuscript text and methods have been updated to reflect this improved process (see Figure 4 and Supplementary Figure 5).

In the revised manuscript we also introduce a new hypothesis for the behavior of *BTG2* since it surfaced as a potential driver in both versions of the analysis. Subsetting the DepMap dependency data to cell lines with amplified *BTG2* shows a similar impact on proximity bias as *MDM4* amplification. Since *BTG2* and *MDM4* are located very close together on chromosome 1, we also looked at all 15 genes with sufficient data between these two genes and found similar behavior with respect to proximity bias (Suppl. Fig. 5). This suggests that with these data it can be difficult to identify specific genes as drivers since copy number changes typically affect larger regions.

For that reason, we also added a section examining results from a gene set enrichment analysis to look for enriched biological processes and found that the strongest enrichments were for "Regulation of Cell Population Proliferation" and "Regulation of Apoptotic Processes".

2. It is apparent in Fig 1D (e.g. chr1), Fig 2a, S2b, that some chromosome arms have inflection points where the likelihood of having a truncation event is dramatically changed. Discussion should be added as to possible causes of this. Data determining whether these sites are at loci containing either haploinsufficient essential genes or other genes that cause gross changes in phenotype would also be useful towards interpretation of these inflection points.

We have added discussion and suggested future lines of research.

We agree that the within-arm structure apparent in the proximity bias heatmaps is suggestive of additional factors influencing this effect, including differences in susceptibility to truncation, epigenetic state influencing CRISPR efficiency, gene haploinsufficiency, gene essentiality, the strength of phenotypic effects caused by genes telomeric from the target loci or some combination of these factors and others. Due to the noisy nature of the proximity bias signal at the individual gene level, it is not possible to reliably attribute these changes to a region smaller than a few dozen genes. Given the

large hypothesis space for potential causative factors, we don't believe that the current data are sufficient to support a conclusion at this time. Moreover, it is clear from figure 1D that the chromosome arm sub-structure is cell type dependent (e.g. chr1p) and so an investigation of these factors would be best served by a larger survey across many cell types with consistent data collection and processing. **We have added discussion to this effect.**

3. In Table 1 and S2, there seems to be a high rate of arm loss centromeric to the cut site as well as telomeric. First, please include whether there is a statistically significant enrichment of telomeric loss (by Fisher's Exact test or similar). Next, in discussion or elsewhere, please include what is the explanation for this. In the discussion you seem to invoke acrocentric chromosomes, however many chromosomes with centromeric loss are not acrocentric. Are genes with loss towards the centromere nearer the centromere than telomere? Relatedly, a discussion of this area would be useful. Are these due to segmental losses as seen in Kosicki et al., 2018? Chromosome bridges as in Leibowitz et al.?

We have updated the manuscript to address these concerns.

We used Fisher's Exact test to test for enrichment of telomeric loss and did find a significant association ($p\text{-value} = 4.9e\text{-}7$); this result has been added to the manuscript. We believe that the increase in telomeric loss is due to a failure of double-strand break repair and loss of genetic material through replication. We also examined if there is any evidence of telomeric loss enrichment in the significantly smaller proportion of genes showing loss in the CRISPRi datasets. Our findings were negative (Fisher's exact test $p\text{-value}$ of 0.31) which is consistent with the hypothesized dependence on double-strand breaks.

We don't believe that the centromeric loss seen here is related to acrocentric chromosomes; rather that previous work (Tsuchida et al. 2023) may have interpreted chromosome *arm* loss as *full-chromosome* loss specifically in chromosome 14 because it is acrocentric. We have

clarified the text on that point. We also tested whether genes exhibiting centromeric loss are nearer the centromere than telomere and did not find a significant association (Mann-Whitney U test p-value 0.65).

Comparing the centromeric loss rates between the CRISPR-Cas9 and CRISPRi data (Table 1) we do see a higher rate of centromeric loss in CRISPR-Cas9, but we don't believe that this limited cross-study data is sufficient to identify root causes in a controlled manner. Given that the CRISPR-Cas9 data are from cancer cell lines, we believe that the most likely cause is the increased susceptibility of deletions near CNVs as reported in Aguirre et al. 2016, Munoz and Amici et al. 2020.

4. The authors of the manuscript should be more careful with language around significance so as to not confuse it with statistical significance. In line 217 there is a suggestion of a

significant change of proximity bias depending upon P53 status. While an effect is apparent in the associated figures, I do not see a test for the statistical significance of the comparison of P53 WT vs LOF. Similar analyses or softening of language is appropriate for other genes as well.

We have revised the manuscript to address these concerns.

Thank you for highlighting this oversight. The line referenced is "We stratified DepMap cell lines by TP53 loss-of-function (LOF) status and found significantly increased proximity bias in a CRISPR map built from putatively TP53-null cell lines compared with one built using only cell lines with putatively functional TP53 (Fig. 4a)." This is a statistical result and p-values have been added to the paper here and in other locations. Language has also been adjusted when statistical significance was not meant to be suggested.

5. More care should be taken to cite papers appropriately. There is no citation to previous work looking at coessentiality on genes (namely Amici et al., 2020), or to work that suggests cell cycle/p53 as a cause for truncation (Cullot et al., 2023). Moreover, although citations are present for the appropriate literature in general, the authors should cite them for their main contributions more clearly. For example, Kosicki et al found kilobase scale deletions, but is not cited for this in the introduction (line 43), Zuccaro et al. (2020) and Leibowitz et al., (2021) both demonstrate chromosome missegregation as the cause for truncation events consistent with your paragraph beginning on line 291.

We have revised the manuscript to address these concerns.

Thank you for the detailed suggestions. We've added citations to Amici and Cullot 2023 in the section on DepMap analyses and updated other citations as suggested here and at the beginning of the review. The review mentioned Cullot et al. 2018; we do now cite Cullot *Nat Comms* 2019 – perhaps the date was a typo?

6. It would be prudent to soften some conclusion language. For example, Add the word “potentially” to line 252 (Potentially outweighed).

We have updated the line suggested and softened the language in several other sections.

7. It is worth finishing the paragraph at line 290 with a statement that no negative consequences due to unintended effects of CRISPR cutting have been seen in patients, potentially mitigating the risk of unintended loss.

A line was added to this effect.

Reviewer 2:

In this manuscript, the authors describe an imaging-based phenotypic profiling approach to characterize the consequences of CRISPR-Cas9 editing. They performed a genome-wide CRISPR-Cas9 screen in a human cancer cell line, using cell imaging as their readout (which they call “phenomics”), and calculated the pairwise similarity of gene KO. They found that genes on the same chromosome-arm exhibit more similar KO profiles (in comparison to random pairs). Transcriptomic data analysis showed loss of chromosome regions telomeric to the cut sites of target genes, suggesting that chromosome truncation may underlie this “proximity bias”. The chromosome-arm proximity bias affected genome-wide CRISPR screening, as gene-gene essentiality relationship was more likely to be detected for genes that reside on the same chromosome-arm (relative to that expected based on databases of known biological relationships). The proximity effect was specific to CRISPR-Cas9 genome editing, and not observed in CRISPRi or shRNA screens. A simple chromosome-arm normalization significantly reduced this effect.

The observation that gene essentiality in CRISPR-Cas9 screens is affected by gene proximity is interesting and of value for the field. However, it’s not quite as novel as presented in the manuscript. The suggestion that this “proximity bias” is driven by chromosome-arm truncations is also intriguing, but the truncations themselves (and their relationship with p53) have been established before, and the specific mechanistic link to the observed “proximity bias” is not sufficiently proven in this manuscript. Overall, while the phenomenon described in the manuscript is certainly interesting, some aspects of the manuscript are not sufficiently novel,

and others are novel but not fully proven by the presented analyses.

Major comments:

1. The manuscript fails to cite/discuss several important and relevant papers, which somewhat diminish the novelty of several aspects of the study.

We have revised the manuscript to address these concerns.

Thank you, we performed an in-depth review of citations and made updates throughout the paper. Any additional suggestions are welcome.

2. To what extent does the “proximity bias” reflect the copy-number effect (described in Aguirre et al. and Meyers et al.)? This may be a major contributor both to the phenotypic similarity that is reported, and (especially) to the observed effect on CRISPR-Cas9 screens, but this is not analyzed and not discussed in this manuscript. While the observation of the “proximity blocks” in karyotypically-normal human primary cells indicates that the effect could not be explained in its entirety by CNAs, CNAs may still be a major contributor to the effect of this “proximity bias” in cancer cell line screens. A thorough analysis is required to separate between these two closely related effects.

We have added a new analysis to address this concern.

To address this, we looked for proximity bias in the DepMap dependency data while controlling for copy number variations by creating a collection of mini-maps across all pairs of chromosome arms (N=741). For each pair of arms:

- We created cosine similarity mini-maps out of just those two arms and calculated arm-level BM probabilities.
- We selected cell lines from the DepMap collection where less than 1% of genes had copy number variations outside of [1.75, 2.25] (number of cell lines min, mean and max of 73, 173.5 and 314 resp.) Supp table 4.
- We constructed cosine similarity mini-maps from just those two arms using only the cell lines with less than 1% CNVs on those arms and calculated arm-level BM probabilities.
- For comparison, we repeated the process using all cell lines in the DepMap RNAi which we know to have no proximity bias.

We find that restricting to cell lines without copy number variants significantly reduces proximity bias, but does not eliminate it with all arm pairs giving probabilities above the baseline of 0.5. This suggests a relationship between copy number variations and proximity bias where regions with more CNVs are prone to increased proximity bias, but the proximity

bias effect is strongly present even in the absence of CNVs with CRISPR-Cas9 cutting. Text has been added to the body of the paper as well as a panel in Supplementary Fig 3.

3. In line with the previous point, the authors report that the chromosome-arm proximity bias was partially corrected in the 23Q2 version of DepMap (Supplementary Fig. 3), which introduced Chronos 2.0 and changed a few technical details in the analysis of data from CRISPR screens. From a practical standpoint, this suggests that the problem has been solved in the DepMap dataset, attenuating the need for the correction proposed at the end of the manuscript. More importantly, can the authors comment on why the 23Q2 changes attenuated the effect so well, and in what ways the correction that they propose is different/better?

We have revised the manuscript to address these concerns.

In the release notes for the 23Q2 DepMap data the authors say the following: (<https://forum.depmap.org/t/announcing-the-23q2-release/2518>) "Additionally, we are aware that there is an artifact in the CRISPR data which causes background correlation in dependency between genes located on the same chromosome arm. To account for this, we've aligned the mean gene effect of each chromosome arm to be the same in every cell line following the original copy number correction. Overall we see an improvement in data quality, as well as a reduction in clustering by chromosome arm in UMAP embeddings." This is essentially the same as the correction we introduce here (using all genes instead of just unexpressed genes on each arm). Though we do not have insight into the inner planning at DepMap, it is notable that this correction was only implemented after the posting of a preprint of this manuscript and engagement with the DepMap team on social media. However, we believe that there is still room for improvement on this correction method for dependency data and the method introduced here is directly applicable to other data modalities (e.g. imaging and transcriptomic data).

4. TP53 status has been shown to affect CRISPR-Cas9 screens (Enache et al. Nat Genet 2020; Bowden et al. eLife 2020, Sinha et al. Nat Commun 2021). The authors report a p53-dependent effect on the "proximity bias". This again might be related to the copy number effect, as TP53-deficient cell lines would carry many more CNAs. And, again, it raises the question of the extent to which the "proximity bias" effect overlaps the previously reported effect of p53 on CRISPR screens. This should be evaluated by a direct comparison.

We believe this concern is addressed by the new analysis described above.

5. The “mechanistic model” for the generation of chromosomal truncations by CRISPR-Cas9 (Fig. 4f) is particularly weak, as it is not supported by substantial data. While the model itself makes sense and fits previous reports from the literature (Nahmad et al. Nat Biotechnol 2022; Tsuchida et al. bioRxiv 2023), the current manuscript provides very little additional support for the model. This part of the Results would fit the Discussion section better than the Results section.

This section of the manuscript has been updated and the figure removed.

We have updated and improved the DepMap proximity bias driver analysis adding new plots, statistical results (Supplementary Table 3) and language. We agree that the diagram presented in Figure 4f was not particularly novel and have decided to remove it from the manuscript.

6. Can the authors compare the imaging-based phenotypes that they identified to those from previous CRISPR screens that used high-content imaging as a readout (e.g., Yan et al. JCB 2021)? If so, does the “proximity bias” exist in those additional datasets as well?

The referenced study is not likely to show the proximity bias effect.

Yan et al. performed several pooled CRISPR screens on a moderate scale (up to 544 genes) with CRISPRa and CRISPRi. Since we believe that the proximity bias effect is dependent on double-strand break induction, we would not expect to see the effect in these data. As we have replicated our findings in two CRISPR-Cas9 imaging datasets and supported them with data from RNAseq and essentiality studies, we don't think that including an analysis of the data from Yan et al. or other CRISPRa/i studies would significantly improve our current work.

Reviewer 3:

The manuscript entitled “High-resolution genome-wide mapping of chromosome-arm-scale truncations induced by CRISPR-Cas9 editing” by Lazar et al reports that unrelated genomically proximal genes located on the same chromosome arm are systematically more similar than expected in CRISPR-Cas9 screening. This phenomena, termed “proximity bias”, was initially discovered using data generated from cellular morphological profiling of two cell lines subjected to CRISPR-Cas9 screening: human umbilical vein endothelial cells by the authors and the U2OS cells by the Joint Undertaking in Morphological Profiling-Cell Painting consortium.

Proximity bias was also found in DepMap (Dependency Map) data. Prior publication and CNV

analysis of the scRNA-seq generated from CRISPR-CAS9 screening indicate that this is likely caused by loss of chromosome arm close to the double-stranded breaks introduced by CAS9 nuclease and had the strongest effect on genes closest to centromere. Proximity bias appears to be affected by cell cycle activity, in particular, TP53 LOF status, which is consistent with prior literature. The authors also showed that proximity bias could be mitigated by geometric correction by subtracting an estimated representation of the chromosome arm based on gene location.

The paper is of tremendous interest to the field as the effect of chromosomal truncation has not been well-appreciated in genetic screening. The study can have the potential influencing the future design of genetic screening and help improving analysis with data generated with such bias. However, there are several major and minor issues related to data analysis and quality that need to be addressed.

1. Novelty on contribution of chromosome truncation/TP53 LOF to proximity bias. This has been reported by Tsuchida et al (“the Mitigation of chromosome loss in clinical CRISPR-Cas9-engineered T cells”, ref 8) and the mechanism is very similar to what’s described in Figure 4d. It is important to emphasize the aspect of replicating the finds made by Tsuchida et al.

We have revised the manuscript to address these concerns.

We have updated the text to emphasize the replication and clarify the comparison with the work from Tsuchida et. al. In particular, that study finds both full and partial loss of the acrocentric chromosome 14, primarily by studying CRISPR-Cas9 targeting at the TRAC locus which is very near the centromere. We hypothesize that there is little selective pressure to retain the centromeric portion of chr14 after a telomeric deletion, so their results are consistent with our findings. Moreover, we expand on that work by studying different cell types with far more loci and several different high-dimensional modalities (imaging, dependency and RNAseq) for measuring the cellular impact of these truncations and their implications for downstream analyses. Additionally, we've removed the model in Figure 4f since it doesn't seem necessary to explain the proposed mechanism.

2. A major concern of the proximity bias is the false positive rate on the dependency genes (target discovery) identified by CRISPR screen due to the proximity effect. Based on the data presented in this paper and the prior publication, chromosome loss only affects a small fraction of the cells (4.2%-15%). Can such a small fraction of cells lead to false discovery of dependency genes or just the general pattern of dependency scores? For example, Online Methods describes that the dependency scores for each gene were centered by subtraction

the means of all cell lines to mitigate the effect of essential genes (lines 457-458). Additional analysis is needed to demonstrate how the proximity bias affects the dependency genes identified in each cell line, in particular those affected by TP53 LOF mutations. The general pattern presented in Figures 3 & 4 does not address this question. Illustration using false positive calls on a well-characterized dependency gene will be highly informative.

We have added a new analysis to address this concern.

We conducted a thorough search of the DepMap data for false positive dependency calls driven by the proximity bias effect within cancer subtypes while controlling for CNV-based effects and found several examples. Briefly, we looked at the 22Q4 DepMap data, grouped cell lines by subtype and tested for differences in dependency values between each subtype and all others but only in cell lines without CNVs. If the telomeric truncation hypothesis is correct, you'd expect to see genes with significant type-specific dependency despite low expression centromeric of known driver genes. In the current version of the manuscript we highlight three such cases in B-lymphoblastic leukemia/lymphoma, Cell Carcinoma Neuroblastoma and Renal Cell Carcinoma (Supplementary Fig. 3e and Supplementary Table 5).

3. Dependency genes were identified by evaluating all cell lines and identify those that are specific to a subset of the cancer. Given proximity effect occurs universally in all cell types, will this (the analytical approaches for dependency genes) mitigate the effect of proximity on dependency gene discovery?

We updated Figure 4 to address this concern.

While we believe that proximity bias will occur in all cell lines given CRISPR-Cas9 cutting, we expect that the prevalence will differ depending on TP53 status and other mitigating factors that vary across cell lines and cancer subtypes. Looking across the full DepMap dependency datasets we detect strong proximity bias signals and stratifying cell lines by the loss or amplification status of particular genes also reveals meaningful relationships between genes and their impact on proximity bias (see updates to Figure 4).

4. Geometric method for proximity bias reduction. The description in Online Methods was too general for evaluation and the resulting code needs to be made publicly available (The URL provided by the authors, <https://github.com/recursionpharma/proxbias>, is not accessible). In addition to improving the correlation, the authors need to show how the adjustment can reduce the potential false positives in the identification of essential genes in CRISPR

screen.

Code has been made public and an analysis added to explore false positive dependencies.

With this resubmission we have made the code public along with several Python notebooks walking through analyses outlined in the paper.

We ran the false positive analysis described in the answer to question 2 on the DepMap 23Q2 data and found that the potential false positives that we identified with the 22Q4 data were reduced. Specifically 5 of the 9 lowly expressed genes centromeric of known drivers in three cancer subtypes were no longer found to be significant. The manuscript was updated (the section on the DepMap proximity bias correction) and statistical details are in Supplementary table 5.

5. RNA-seq analysis with the results presented in Table 1, Supplementary Table S2 and Figure 2 D-F.

The manuscript has been updated to clarify this analysis.

- a. The narrative on line 146-151 suggested 150 target genes flanking the target site were evaluated while the data presented in Table 1 is <150X2 genes. Please clarify this.

Table 1 does not show the genes that are lost as a result of a perturbation, but the targeted genes that result in a substantial loss in 150 genes either 5' or 3' from the target. The number of genes in Table 1 is only expected to be smaller than 2 times the number of targeted genes and it does not have any relation to the neighborhood window size during evaluation.

- b. The proposed model indicating chromosome-arm level changes; however the results showed that only 8%-33% of targeted genes exhibited copy number loss, which is a surprise. Please clarify

In our evaluation, we consider a gene perturbation to have a loss if >70% of the 150 genes in its 5' or 3' neighborhood exhibit reduced copy number and find that between 8% and 33% of genes meet this criteria. A number of factors likely influence whether a CRISPR-Cas9-induced double strand break might result in a deletion including chromatin state, guide targeting efficiency and survival advantages.

- c. Supplementary Table S2 also shows a high variability in % cells affect by copy

number loss in the target genes. If the target genes are all within the chromosomal loss, we would expect to see consistent number of cells across. Please clarify the results. By contrast, CNV loss shown in Figure 2 looks uniform.

Related to the points above, the genes presented in Table 1 are not those in the deleted region, but the targeted genes that exhibit signs of a deletion nearby in the sequencing data. Supplementary Table 2 reports the percentage of cells that lose 70% of reads for 150 neighboring genes when the target gene is perturbed. Figure 2 shows only cells (one row per cell) with losses near the target genes (row labels) in a 150-gene neighborhood as a result of the perturbation (CRISPR-Cas9 editing). Figure legends were updated to clarify.

- d. According to Online Methods “Analysis of public scRNAseq data”, CNV analysis from scRNA-seq does not appear to be equivalent to target gene analysis, please clarify if that is the case. Was the data presented in Figure 2d and 2e based on inferCNV or it is based on the analysis described in Online Methods “An Analysis of public scRNAseq data”?

The data presented in Figure 2d and 2e are based both on InferCNV and the analysis described in the Online Methods. The output from InferCNV is an estimate for copy number at each gene locus. We used those values to determine which genes exhibit consistent blocks of loss near the target gene (150-gene neighborhoods on both the 5' and 3' regions). We rewrote the description of the scPerturb analysis results and the associated online methods to clarify the above points.

6. There are three different versions of DepMap data used, 19Q3, 22Q4 and 23Q2. Please describe the rationale for using these different versions of data.

We clarified the purpose of each data version in the manuscript.

19Q3 was used because it powers the ShinyDepMap gene-gene relationship app which is a commonly used resource and explored for Supplementary Fig. 3c,d. 22Q4 was used because it has more cell lines and an updated processing pipeline including a correction for copy-number dependencies (Amici et al. 2020). 23Q2 was released after we posted a preprint of this manuscript and introduced a correction very similar to our methods. The difference is that they center each arm by the mean of all genes on that arm instead of only unexpressed genes. We show results from that dataset as an analogue to our correction applied to the imaging data.

Reviewer 3: Minor issues

1. Refs 1-3 do not appear to be relevant to chromosome truncations caused by CRISPR,

which was the original intent. Instead, these appear to describe the clinical application of genetic engineering. Please make corrections by citing the appropriate references.

We did a thorough review of all citations and updated many references. Please let us know if there are changes that you think are still needed.

2. The content of Table 1 is duplicated in Supplementary Table 2.

We have updated this file.

3. The section of Online Methods “An Analysis of public scRNAseq data” describes both Table 1 and 2 in Line 433, however, Table 2 can not be found. Does this refer to Supplementary Table 2?

We regret the error, a previous version of the manuscript separated Table 1 into two parts. This has been corrected.

4. Figure 2 legend. The description of the CNV maps in panels d-f is mixed with that of panel C, a schematic drawing for the correlation model, which is quite confusing.

Thank you for calling attention to this, we have updated the description.

5. Line 843: 22Q2 should be relabeled as 23Q2

Thank you for calling attention to this, we have fixed this typo.

6. Line 179-182: Supplementary Figure S3 showed that DepMap 23Q2 has mitigated the effect of proximity bias by “by a correction similar to that described below”. Is this a speculation? If not, please verify with the DepMap development team on how they reduced this effect in their new data release.

The DepMap release notes say the following:

"Additionally, we are aware that there is an artifact in the CRISPR data which causes background correlation in dependency between genes located on the same chromosome arm. To account for this, we've aligned the mean gene effect of each chromosome arm to be the same in every cell line following the original copy number correction. Overall we see an improvement in data quality, as well as a reduction in clustering by chromosome arm in UMAP embeddings." (<https://forum.depmap.org/t/announcing-the-23q2-release/2518>) This is nearly equivalent to our correction, but uses all genes instead of just the unexpressed

(since expression differs across cell lines) and was only implemented after the DepMap team engaged with our preprint of this work on social media.

Decision Letter, first revision:

12th Jan 2024

Dear Dr Haque,

Your Article, "High-resolution genome-wide mapping of chromosome-arm-scale truncations induced by CRISPR-Cas9 editing" has now been seen by 3 referees. You will see from their comments below that while they find your work of interest, some remaining concerns are raised. We are interested in the possibility of publishing your study in Nature Genetics, but would like to consider your response to these concerns in the form of a revised manuscript before we make a final decision on publication.

Reviewers are satisfied with how the manuscript has improved during revision however, Reviewers #1 and #2 require additional analyses that prove that proximity and CNV effects are sufficiently independent and have a few suggestions to improve this. Reviewer #3 has remaining minor points. Please do not hesitate to get in touch if you would like to discuss these issues further.

We therefore invite you to revise your manuscript taking into account all reviewer and editor comments. Please highlight all changes in the manuscript text file. At this stage we will need you to upload a copy of the manuscript in MS Word .docx or similar editable format.

*2) If you have not done so already please begin to revise your manuscript so that it conforms to our Article format instructions, available here.

*3) Include a revised version of any required Reporting Summary:

Please be aware of our guidelines on digital image standards.

[redacted]

We hope to receive your revised manuscript within four to eight weeks. If you cannot send it within this time, please let us know.

Nature Genetics is committed to improving transparency in authorship. As part of our efforts in this direction, we are now requesting that all authors identified as 'corresponding author' on published papers create and link their Open Researcher and Contributor Identifier (ORCID) with their account on the Manuscript Tracking System (MTS), prior to acceptance. ORCID helps the scientific community achieve unambiguous attribution of all scholarly contributions. You can create and link your ORCID from the home page of the MTS by clicking on 'Modify my Springer Nature account'. For more information please visit please visit www.springernature.com/orcid.

Sincerely,
Chiara

Chiara Anania, PhD
Associate Editor
Nature Genetics
<https://orcid.org/0000-0003-1549-4157>

Referee expertise:

Referee #1:

Referee #2:

Referee #3:

Reviewers' Comments:

Reviewer #1:

Remarks to the Author:

The revised manuscript by Lazar and colleagues is improved. It has included new analyses towards establishing that proximity bias is distinct from copy-number effect. Using DepMap it provides a necessary and updated analysis of genotypes that might be permissive or preventative of proximity bias, including fixing an error from the previous approach used in the initial submission. It has also added helpful context and discussion of similar work in the field, and for interpretation of results. Although the DepMap has made a similar proximity bias correction to the one applied in the manuscript, it is not widely publicized, and therefore, in my opinion, does not diminish novelty. I continue to believe the manuscript will be widely read and is worthy of publication in Nature Genetics. Prior to publication, I recommend the following changes, largely to improve differentiation between proximity bias and copy-number effect:

(1) Lines 193-210: Confirm, and if confirmed, add a statement that the gRNAs showing proximity bias in copy number neutral chromosome arms only align to single targets (for example, not repetitive elements).

(2) For the analysis surrounding Fig S3e (lines 222-237):

a. I find the figure rather unintelligible as to the patterns being observed. This figure can simply be removed while maintaining the analysis in the text.

b. The methods should be updated to describe how the authors define "low expression"

c. Key to showing that the correction removes false positives, Supplementary Table 5 should be updated to also contain analysis after either your geometric correction (preferred), or the DepMap correction. This will additionally make the statement in lines 325-326 more accurate.

d. Do the gRNAs targeting the highlighted genes (eg C2orf73) align to small repetitive elements or have high rates of off-target sites with only one or a few mismatches? If not, please include a sentence verifying they do not. If they do, then this would support copy-number effect for these guides, even if the regions they are in, as a whole, are single-copy.

(3) Lines 283-284: I greatly appreciate the new and improved analysis performed in this paragraph. However, as highlighted in the paragraph starting on line 285, it can be difficult to define causative genes due to chromosome-position effects. Here, chromosome-position effects might also explain the result that MDM4 amplification in TP53 LOF backgrounds decreases proximity bias. For this reason, and lacking a clear hypothesis explaining the result, I recommend the authors remove speculation that "MDM4 may have an additional role".

Reviewer #2:

Remarks to the Author:

The authors have generally addressed my comments and concerns.

As mentioned in my original review, my opinion is that the novelty of this manuscript is borderline for Nature Genetics.

The finding that the proximity effect is partially independent of the CNV effect is important, but I'm not convinced that these two effects were optimally separated in the approach that the authors have taken. Especially given that the authors' attempt to correct for CNV did reduce the proximity bias to a considerable degree, I would prefer seeing additional analyses that are trying to tease apart these two effects (perhaps assessing the CN effect and using it as a covariate in proximity bias analyses?).

Reviewer #3:

Remarks to the Author:

The authors have addressed all the issues raised in the prior review. Given a correction to proximity bias can be easily implemented as described in the section Methods "Geometric method for proximity bias reduction", the authors may consider adding the following information to improve clarity: 1) specify whether the "unexpressed" genes were defined using the gene expression data derived from cell lines prior to CAS9 treatment; 2) specify the meaning of vector when referring to "subtraction of the mean vector for unexpressed genes on a specific chromosome arm".

Author Rebuttal, first revision:

Reviewer Responses

Reviewer #1:

Remarks to the Author:

The revised manuscript by Lazar and colleagues is improved. It has included new analyses towards establishing that proximity bias is distinct from copy-number effect. Using DepMap it provides a necessary and updated analysis of genotypes that might be permissive or preventative of proximity bias, including fixing an error from the previous approach used in the initial submission. It has also added helpful context and discussion of similar work in the field, and for interpretation of results. Although the DepMap has made a similar proximity bias correction to the one applied in the manuscript, it is not widely publicized, and therefore, in my opinion, does not diminish novelty. I continue to believe the manuscript will be widely read and is worthy of publication in Nature Genetics. Prior to publication, I recommend the following changes, largely to improve differentiation between proximity bias and copy-number effect:

1. Lines 193-210: Confirm, and if confirmed, add a statement that the gRNAs showing proximity bias in copy number neutral chromosome arms only align to single targets (for example, not repetitive elements).

Yes, these gRNAs only align to single targets. We have updated the manuscript to highlight this point.

Details:

DepMap provides the following files on guide mapping at <https://depmap.org/portal/download/>
22Q4 data: KYGuideMap.csv, AvanaGuideMap.csv

23Q2 data: KYGuideMap.csv, AvanaGuideMap.csv

23Q4 data: KYGuideMap.csv, HumagneGuideMap.csv, AvanaGuideMap.csv

They remove guides that map to multiple loci in their Chronos pipeline (applied to datasets starting with 22Q4) and we only work with data after this pre-processing. A sample of one of these files is pictured below. We have updated the manuscript to address this concern.

KYGuideMap

sgRNA	GenomeAlignment	Gene	nAlignments	DropReason	UsedByChronos
AAAAAAAAATCCAGAACCT	chr15_45432096_+	C15orf48 (84419)	1.0		TRUE
AAAAAAAAATATGCCCGTGGA	chr12_8129357_+	CLEC4A (50856)	1.0		TRUE
AAAAAAGCATTTAGGCAGG	chr14_89931586_-	EFCAB11 (90141)	1.0		TRUE
AAAAAAGCTTGCATTAGAC	chr5_69195786_+	CENPH (64946)	1.0		TRUE
AAAAAATATCGTGTCAAGT	chr1_193049770_+	UCHL5 (51377)	1.0		TRUE
AAAAAATCAGCCACGCGAC	chr11_18173799_-	MRGPRX4 (117196)	1.0		TRUE
AAAAAATGAGAGCTGAAGA	chrX_33041522_+	INTERGENIC_chrX	3.0	aligns with multiple genes	FALSE
AAAAAATGAGAGCTGAAGA	chr4_98909732_-	EIF4E (1977)	3.0	aligns with multiple genes	FALSE
AAAAAATGAGAGCTGAAGA	chr5_77708272_-	TBCA (6902)	3.0	aligns with multiple genes	FALSE
AAAAAATGTCAGTCGAGTG	chr13_99255303_+	GPR18 (2841)	1.0		TRUE
AAAAAATGTTTCATTGTG	chr6_53294382_-	ELOVL5 (60481)	1.0		TRUE

2. For the analysis surrounding Fig S3e (lines 222-237):

a. I find the figure rather unintelligible as to the patterns being observed. This figure can simply be removed while maintaining the analysis in the text.

Thank you for the perspective, the figure has been removed and main and supplemental text updated.

b. The methods should be updated to describe how the authors define “low expression”

The expression values in transcripts per million reads (TPM) are given in Supplementary Table 5. All examples given in the manuscript show expression below 0.3 TPM and significantly higher essentiality in the given cancer subtype (Benjamini-Hochberg adjusted t-test p-value < 0.01). The text has been updated to clarify and a column has been added to Supplementary Table 5 highlighting which genes in which cell lines are defined as having low expression.

c. Key to showing that the correction removes false positives, Supplementary Table 5 should be updated to also contain analysis after either your geometric

correction (preferred), or the DepMap correction. This will additionally make the statement in lines 325-326 more accurate.

Supplementary table 5 includes results from two versions of the DepMap data. The 22Q4 data has no proximity bias correction applied, whereas in 23Q2 a correction as similar as possible to our method has been applied. The correction applied by the DepMap team in the process of building the 23Q2 release necessarily differs from that presented in our manuscript because there are different expression values for each of the cell lines, so defining a set of unexpressed genes per chromosome arm is not possible.

We have clarified this in the text and added text to the mentioned statement highlighting the reduction in potential false positive dependencies when the correction is applied.

- d. Do the gRNAs targeting the highlighted genes (eg C2orf73) align to small repetitive elements or have high rates of off-target sites with only one or a few mismatches? If not, please include a sentence verifying they do not. If they do, then this would support copy-number effect for these guides, even if the regions they are in, as a whole, are single-copy.

As mentioned above, the DepMap Chronos processing pipeline (used for both the 22Q4 and 23Q2 datasets) excludes guides that map to multiple genomic locations.

3. Lines 283-284: I greatly appreciate the new and improved analysis performed in this paragraph. However, as highlighted in the paragraph starting on line 285, it can be difficult to define causative genes due to chromosome-position effects. Here, chromosome-position effects might also explain the result that MDM4 amplification in TP53 LOF backgrounds decreases proximity bias. For this reason, and lacking a clear hypothesis explaining the result, I recommend the authors remove speculation that “MDM4 may have an additional role”.

Thank you for the thoughtful suggestion, the speculative line has been removed.

Reviewer #2:

Remarks to the Author:

The authors have generally addressed my comments and concerns.

As mentioned in my original review, my opinion is that the novelty of this manuscript is borderline for Nature Genetics.

The finding that the proximity effect is partially independent of the CNV effect is important, but I'm not convinced that these two effects were optimally separated in the approach that the authors have taken. Especially given that the authors' attempt to correct for CNV did reduce the proximity bias to a considerable degree, I would prefer seeing additional analyses that are trying to tease apart these two effects (perhaps assessing the CN effect and using it as a covariate in proximity bias analyses?).

Please note that proximity bias is observed in karyotypically-normal primary HUVEC cells (Figure 1d/e/f above the diagonal and quantified in Figure 1g); this observation demonstrates that CNV cannot be the sole cause of the observed effects, and that proximity bias can significantly confound results even in screens of euploid cells.

It is indeed more challenging to separate these effects in the DepMap data because of the high rates of CNV in the cell lines used, and the heterogeneity of these lines. Nevertheless, we believe this data also demonstrates the independence of CNV and proximity bias:

- Please note that the DepMap shRNA data (Suppl. Fig 4a, below the diagonal) shows no proximity bias effects, despite being built on cell lines with significant CNV.
- The analysis in Suppl Fig 3b (computing Brunner-Munzel probabilities on subsets of cell lines without CNV in the tested chromosome arms) suffers from deflation of its test statistic and reduction in power as the number of tested cell lines decreases, which can present the appearance of reduced proximity bias. To address this, we have added a data series to Suppl Fig 3b that matches the test set sizes for the CNV-filtered comparisons, but without the CNV filter (that is, modeling only the test statistic deflation).

Ultimately, our goal with this manuscript is to present evidence for a novel CRISPR-related bias, and we believe the evidence clearly demonstrates that while there may be *interaction* between proximity bias and CNV, that the effects we report cannot be sufficiently explained by CNV alone. While it is interesting future work to more optimally decouple proximity bias and CNV, it is beyond the scope of this work (and we are not certain that the DepMap dataset alone would be sufficient for that future work).

Reviewer #3:

Remarks to the Author:

The authors have addressed all the issues raised in the prior review. Given a correction to

proximity bias can be easily implemented as described in the section Methods "Geometric method for proximity bias reduction", the authors may consider adding the following information to improve clarity: 1) specify whether the "unexpressed" genes were defined using the gene expression data derived from cell lines prior to CAS9 treatment; 2) specific the meaning of vector when referring to "subtraction of the mean vector for unexpressed genes on a specific chromosome arm".

Thank you for the suggestions, we have updated the manuscript accordingly.

Decision Letter, second revision:

7th Feb 2024

Dear Dr. Haque,

Thank you for submitting your revised manuscript "High-resolution genome-wide mapping of chromosome-arm-scale truncations induced by CRISPR-Cas9 editing" (NG-A62901R1). It has now been seen by the original referees and their comments are below. The reviewers find that the paper has improved in revision, and therefore we'll be happy in principle to publish it in Nature Genetics, pending minor revisions to satisfy the referees' final requests and to comply with our editorial and formatting guidelines.

Sincerely,
Chiara

Chiara Anania, PhD
Associate Editor
Nature Genetics
<https://orcid.org/0000-0003-1549-4157>

Reviewer #1 (Remarks to the Author):

The authors have addressed my concerns with appropriate text and I have nothing more to add.

Reviewer #2 (Remarks to the Author):

I congratulate the authors for an interesting study.
I have no further comments on this manuscript at this stage.

Final Decision Letter:

18th Apr 2024

Dear Dr. Haque,

I am delighted to say that your manuscript "High-resolution genome-wide mapping of chromosome-arm-scale truncations induced by CRISPR-Cas9 editing" has been accepted for publication in an upcoming issue of Nature Genetics.

Your paper will be published online after we receive your corrections and will appear in print in the next available issue. You can find out your date of online publication by contacting the Nature Press Office (press@nature.com) after sending your e-proof corrections.

Before your paper is published online, we shall be distributing a press release to news organizations

worldwide, which may very well include details of your work. We are happy for your institution or funding agency to prepare its own press release, but it must mention the embargo date and Nature Genetics. Our Press Office may contact you closer to the time of publication, but if you or your Press Office have any enquiries in the meantime, please contact press@nature.com.

Please note that *Nature Genetics* is a Transformative Journal (TJ). Authors may publish their research with us through the traditional subscription access route or make their paper immediately open access through payment of an article-processing charge (APC). Authors will not be required to make a final decision about access to their article until it has been accepted. Find out more about Transformative Journals

Authors may need to take specific actions to achieve compliance with funder and institutional open access mandates. If your research is supported by a funder that requires immediate open access (e.g. according to Plan S principles) then you should select the gold OA route, and we will direct you to the compliant route where possible. For authors selecting the subscription publication route, the journal's standard licensing terms will need to be accepted, including [a href="https://www.nature.com/nature-portfolio/editorial-policies/self-archiving-and-license-to-publish"](https://www.nature.com/nature-portfolio/editorial-policies/self-archiving-and-license-to-publish). Those licensing terms will supersede any other terms that the author or any third party may assert apply to any version of the manuscript.

If you have not already done so, we invite you to upload the step-by-step protocols used in this

manuscript to the Protocols Exchange, part of our on-line web resource, natureprotocols.com. If you complete the upload by the time you receive your manuscript proofs, we can insert links in your article that lead directly to the protocol details. Your protocol will be made freely available upon publication of your paper. By participating in natureprotocols.com, you are enabling researchers to more readily reproduce or adapt the methodology you use. Natureprotocols.com is fully searchable, providing your protocols and paper with increased utility and visibility. Please submit your protocol to <https://protocolexchange.researchsquare.com/>. After entering your nature.com username and password you will need to enter your manuscript number (NG-A62901R2). Further information can be found at <https://www.nature.com/nature-portfolio/editorial-policies/reporting-standards#protocols>

Sincerely,
Chiara

Chiara Anania, PhD
Associate Editor
Nature Genetics
<https://orcid.org/0000-0003-1549-4157>